# Retrievals of Precipitable Water Vapor and Aerosol Optical Depth from direct sun measurements with EKO MS711 and MS712 Spectroradiometers

Congcong Qiao[1,2], Song Liu[1,2], Juan Huo[1], Xihan Mu[3], Ping Wang[4], Shengjie Jia[5], Xuehua Fan[1], Minzheng Duan[1,2*]

[1]LAGEO, Institute of Atmospheric Physics, Chinese Academy of Sciences, Beijing, 100029, China
[2]College of Earth and Planetary Sciences, University of Chinese Academy of Sciences, Beijing, 100049, China
[3]State Key Laboratory of Remote Sensing Science, Beijing Normal University, Beijing, 100875, China
[4] Royal Netherlands Meteorological Institute (KNMI), De Bilt, the Netherlands
[5]Beijing Keytec Technology Co., Ltd., Beijing, 100102, China

*Correspondence to*: Minzheng Duan (dmz@mail.iap.ac.cn)

**Abstract.** Based on the strict radiative transfer algorithm, a new method is developed to derive the Precipitable Water Vapor (PWV) and Aerosol optical depth (AOD) from the ground-based direct sun irradiance measurements. The attenuated direct irradiance from 300 nm to 1700 nm was measured by a pair of grating spectroradiometers MS711 and MS712 produced by EKO INSTRUMENTS, located at the Institute of Atmospheric Physics (IAP), Chinese Academy of Sciences (39.98° N, 116.38° E), from June 2020 to March 2021. Compared with regular sun photometers such as CE-318 and POM, EKO instruments can measure a wider range of continuous spectra, but their Field Of View (FOV) is also relatively large. In the PWV inversion of this work, a strong water vapor absorption band around 1370 nm is introduced to retrieve PWV in a relatively dry atmosphere. The CircumSolar Radiation (CSR) of the EKO instruments is corrected to reduce the influence of scattering from relatively larger FOV on the AOD inversion. The PWV and AOD inversion results obtained by MS711 and MS712 are compared with the synchronous data of the CE-318 sun photometer. The two retrieval results are highly consistent. The correlation coefficient, mean bias, and standard deviation of $PWV_{EKO}$ and $PWV_{CIMEL}$ are 0.999, -0.027 cm (-2.42 %), and 0.054 cm (3.93 %) respectively, and the relative deviations of the differences between the two are slightly larger for drier air (PWV<0.5 cm) and lower solar elevation angle. The correlation coefficients of $AOD_{EKO}$ and $AOD_{CIMEL}$ at 380, 440, 500, 675, 870, and 1020 nm are greater than 0.99, and the relative deviations vary between -6.59 % and 4.27 %.

## 1 Introduction

Water vapor and aerosols are two key components of the atmosphere (Bojinski et al., 2014; IPCC, 2013), and the current accuracy of their indirect measurements from spaceborne instruments (Dubovik et al., 2019; Kaufman et al., 2002; Kokhanovsky, 2013) is unsatisfactory in the evaluation of earth climate simulations and environment modelling (IPCC, 2021), often needing to be combined with ground-based measurements for higher accuracy retrievals (Li et al., 2019; WMO, 2016).

As for PWV, ground observation methods include Global Positioning System (GPS), MicroWave radiation Profiler System (MWPS), sun photometers (CE-318, POM, MFR), and others. GPS signals delayed by the atmosphere can be used to obtain global PWV at a relatively high temporal resolution, but the algorithm still needs to be improved for accuracy (Bevis et al., 1992; Wang et al., 2007). MWPS measures the microwave radiation emitted from the atmosphere and yields the vertical water

vapor profile, which can then be integrated to derive PWV (Güldner and Spänkuch, 2001; J. and Güldner, 2013). The advantage of using the microwave for PWV is that aerosols have little effect, but the disadvantage is that this kind of instrument is generally very expensive. Sun photometers are easy to operate, and it is economical to build the observation network with them (Augustine et al., 2008; Wehrli, 2003), so they are widely used to monitor water vapor and aerosols (Barreto et al., 2014; Cuevas Agulló et al., 2015; Kazadzis et al., 2014; Schmid et al., 1999). Among them, the CE-318 produced by French CIMEL

corporate is the most popular one and used in the Aerosol RObotic NETwork (AERONET) project (Holben et al., 1998), China Aerosol Remote Sensing Network (CARSNET) (Che et al., 2016), and Sun–Sky Radiometer Observation Network (SONET) (Li et al., 2018). Similar instruments such as POM are deployed in the SKY radiometer NETwork (SKYNET) (Campanelli et al., 2012; Campanelli et al., 2014).

Currently, AERONET is the most recognized ground-based aerosol observation network. Since the 1990s, NASA and

PHOTONS (PHOtométrie pour le Traitement Opérationnel de Normalisation Satellitaire) have established more than 500 sites worldwide based on the CE-318 sun photometer, which could provide water vapor and aerosol optical properties through the measurements in the visible and short-wave infrared bands. The aerosol and PWV products derived from CE-318 are often used as references to validate those obtained by other methods. Additionally, some scientists have attempted to retrieve PWV and AOD using spectral measurements. Estellés et al. (2006) used li-COR 1800 spectroradiometer to retrieve AOD, their

results showed differences with those from CE-318 of 0.01-0.03 and 0.02-0.05 in the ultraviolet and visible bands, respectively. Cachorro et al. (2009) compared AOD obtained by li-COR and sun photometer and found differences of AOD within 0.02 in the spectral range of 440-1200 nm. The results of PWV and AOD from spectral measurements of Precision Solar spectroRadiometer (PSR) at Meteorologisches Observatorium Lindenberg – Richard Assmann Observatorium (MOL–RAO) showed a standard deviation of 0.18 cm for PWV and an overestimation of 0.01-0.03 for AOD at visible and near-infrared

wavelengths compared to CE-318. The PWV given by the monochromatic method around 940 nm has great variability at different wavelengths (Kazadzis et al., 2018a; Kazadzis et al., 2018b; Kazadzis et al., 2014; Raptis et al., 2018). García et al. (2020;2021) retrieved PWV and AOD using the EKO MS711 spectroradiometer at Izana Observatory in Spain and compared them with CE-318, showing that PWV has a mean bias of 0.033 cm, and the AOD is broadly in line.

A method of simple Lambert-Beer law was used to retrieve AOD and a three-parameter formula proposed by Ingold et al.

(2000) was used to retrieve PWV with measurements of the 940 nm water vapor band in the above-mentioned publications. Since the three-parameter method is very sensitive to the instrument slit function, air mass, and wavelength, a spectral fitting algorithm is proposed to derive the PWV. In this work, Direct Normal solar Irradiance (DNI) at 300-1700 nm was measured with EKO MS711 and MS712 spectroradiometers, then AOD and PWV were retrieved and compared to those of CE-318. In addition, the water vapor absorption band near 1370 nm is introduced to retrieve PWV, which is more sensitive to water vapor,

and the signal is not easily measured when the water vapor content is high, but it is expected to improve the water vapor retrieval efficiency in dry environments.

## 2 Instruments

The grating spectroradiometers MS711 and MS712 are designed and developed by EKO INSTRUMENTS and can be used to measure the attenuation of direct solar beams in the range of 300-1700 nm, with a high time resolution of 1 minute. The Full
Width at Half Maximum (FWHM), wavelength accuracy, full Field Of View (FOV) angle and exposure time of the two spectroradiometers are the same, in order of < 7 nm, ±0.2 nm, 5°, and 10-5000 ms. The differences between the two are that the average wavelength interval is 0.4 nm and 2.0 nm, respectively, and the temperature control is 25±2 ℃ and -5±0.5 ℃, respectively. The main specifications related to MS711 and MS712 are listed in Table 1.

**Table 1 EKO MS711 and MS712 spectroradiometers specifications**

| Sensor | MS711 | MS712 |
|---|---|---|
| Wavelength | 300-1100 nm | 900-1700 nm |
| Wavelength Interval | 0.3-0.5 nm | 1.2-2.2 nm |
| Temperature Control | 25±2 ℃ | -5±0.5 ℃ |
| Dome material | Synthetic Quartz | BK7 |
| Operating conditions | Tem: 0~+40 ℃, Humidity: 0~90 %RH*No condensation | |
| Spectral Resolution | <7 nm | |
| Wavelength Accuracy | ±0.2 nm | |
| Exposure Time | 10-5000 ms | |
| Communication | RS-422 / 232C | |
| Power supply | 100-240 VAC, 50/60 Hz | |
| Field of view (FOV) | 5° | |

CE-318 is a narrow-band sun photometer developed by CIMEL Electronique in France, which can directly measure the radiance of the sun and the sky. Measurements are usually made every 10-15 minutes at 340, 380, 440, 500, 675, 870, 940, 1020, and 1640 nm by rotating filter wheels. The bandwidth of the instrument is 2 nm and 4 nm at 340 nm and 380 nm
respectively, and 10 nm in other bands (Schmid et al., 1999). The FOV of CE-318 is about 1.2° and is calibrated annually.

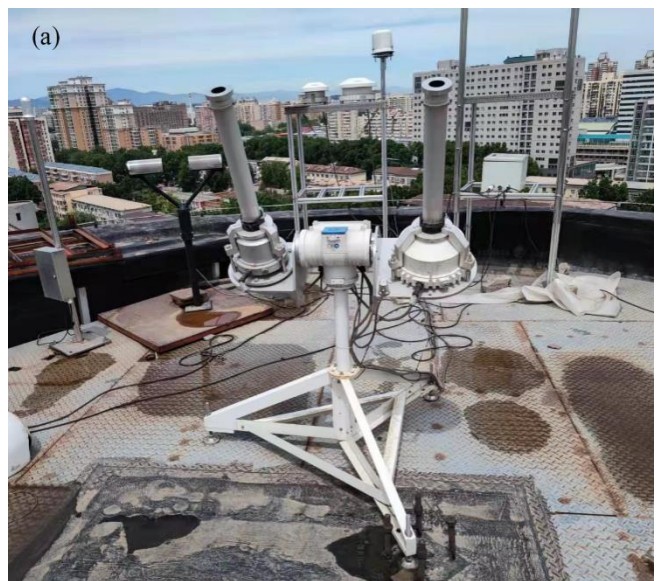
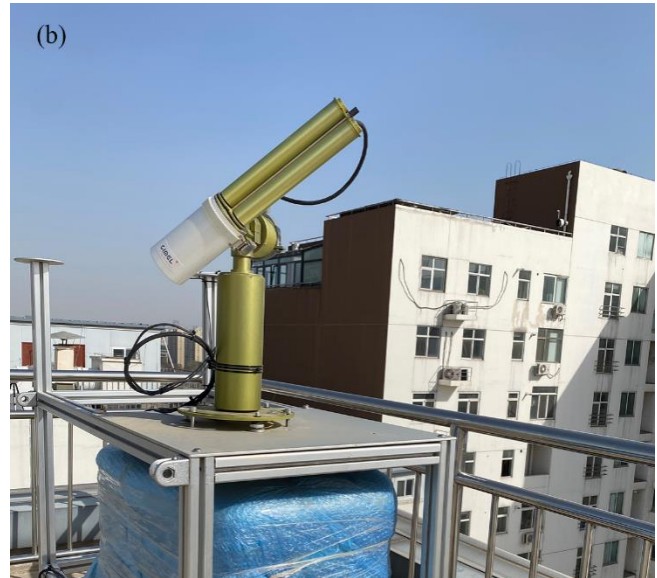

**Figure 1. The EKO spectroradiometers (a) and CE-318 sun photometer (b) are collocated at the top of IAP's building.**

The instruments are collocated in the Institute of Atmospheric Physics (IAP), Chinese Academy of Sciences (CAS), Beijing (39.98° N, 116.38° E, 92 m a.s.l, Fig. 1), located in a relatively dry area in northern China, where most precipitation occurs in
summer, and the water vapor content in the atmosphere of other seasons is very low. The data used here are collected from June 2020 to March 2021, and level 1.5 data of AERONET (https://AERONET.gsfc.nasa.gov/) are used for comparison.

## 3 Inversion Method

### 3.1 Cloud screening

Cloud contamination needs to be avoided before performing the inversion. Consider that the change of clouds in a short time
is usually more drastic than that of aerosols, the temporal resolution of EKO measurements is relatively high at 1 min. We referred to the methods proposed by Smirnov et al. (2000) and Michalsky et al. (2001) for cloud screening of ground-based measurements by imposing a threshold on the standard deviation of the measurements to extract the clear-sky portion of the dataset. Specifically, in order to implement cloud detection, if the standard deviation of the measured value of MS711 at 870 nm within 5 minutes is greater than $15\ w \cdot m^2 \cdot \mu m^{-1}$, and the standard deviation of the measured value of MS712 at 1370 nm
within 5 minutes is greater than $1\ w \cdot m^2 \cdot \mu m^{-1}$, we label it as cloud contaminated.

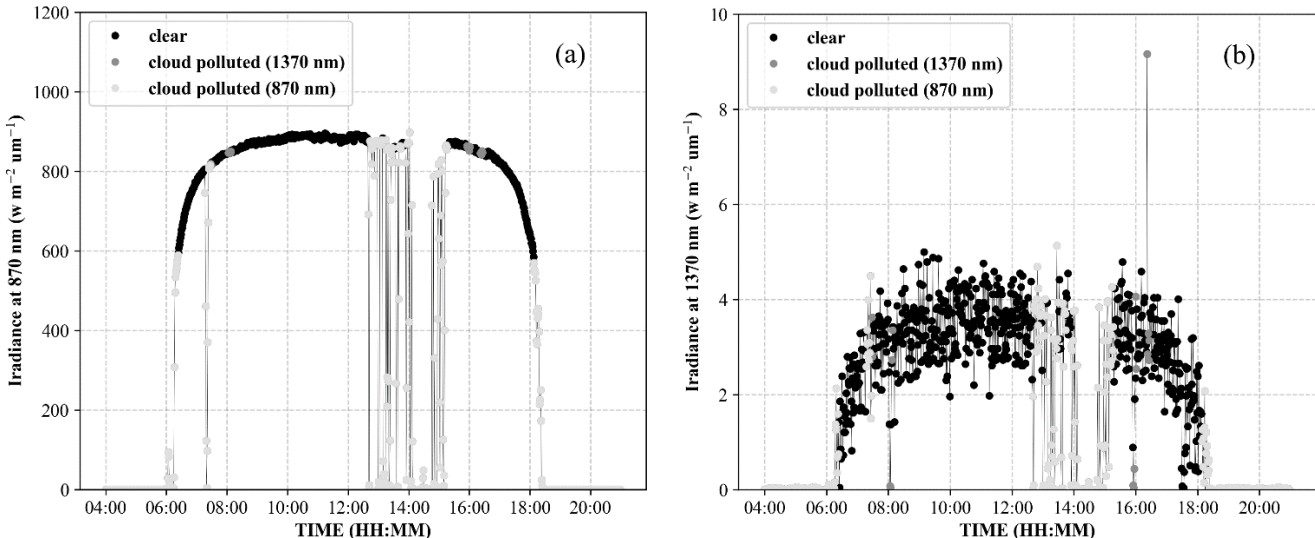

**Figure 2. Direct normal irradiance measurements of EKO instruments at 870 nm (a) and 1370 nm (b) on September 8, 2020, at IAP. Cloudy parts and very small measurements are shown in grey, light grey and dark grey are filtered out using 870nm and 1370nm measurements respectively, and clear-sky parts are shown in black.**

Figure 2 (a) and (b) represent the diurnal variations of the radiation measurements of MS711 at 870 nm and MS712 at 1370 nm on September 8, 2020, respectively. As can be seen from the figure, the cloud screening effect of this method is quite good, but the current threshold is manually selected, which cannot completely exclude missed or false detection.

### 3.2 PWV inversion

Figure 3 shows the theoretical transmittance curves for Rayleigh scattering, aerosols, and water vapor from 300 nm to 1700 nm calculated by MODTRAN 4.3 (Larar et al., 1999) at 0° solar zenith angle. WMO (2005) recommends the use of 719, 817, and 946 nm central wavelengths to obtain PWV, which are marked with the grey arrows in Fig. 3. Ingold et al. (2000) compared the water vapor inversion results of these wavelengths and found that 946 nm is of the most suitable for PWV retrieval. The water vapor data provided by the CE-318 sun photometer are also obtained by the band near 946 nm (Smirnov et al., 2004). However, as demonstrated in Fig. 3, the transmittance at 946 nm turns to be less sensitive to water vapor as the air becomes drier, while the water vapor absorption remains strong around 1370 nm, therefore, the water vapor absorption window of 1350-1450 nm was considered for PWV inversion in very dry atmospheres.

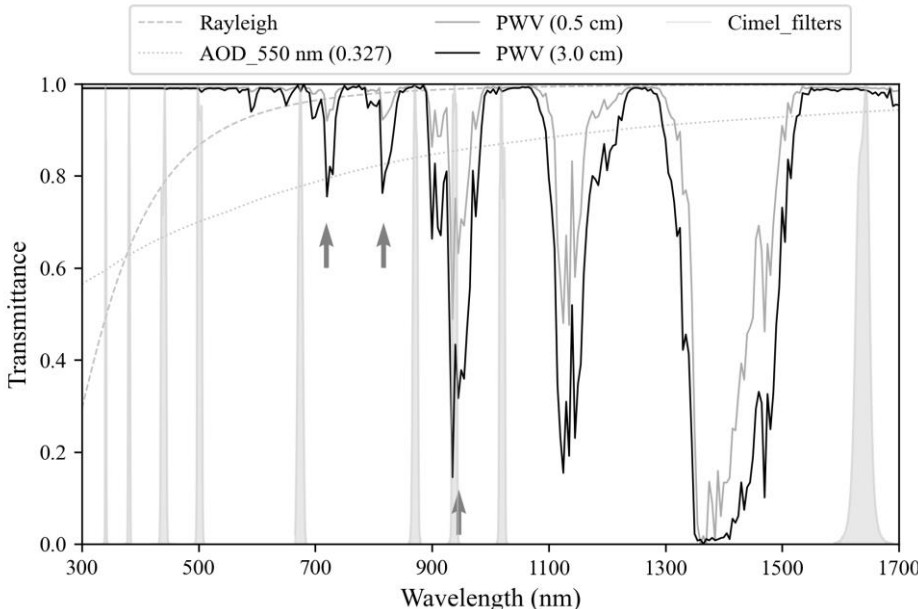

**Figure 3. The spectrum response curves of CE-318 photometer's filter wheels, and the transmittance of water vapor, aerosols, and Rayleigh scattering in the spectral range of 300–1700 nm, which are calculated by MODTRAN 4.3 at SZA=0°, PWV=0.5 cm, PWV=3.0 cm, and Boundary Aerosol Model=Rural extinction (spring-summer), VIS=23 km. The wavelengths pointed by the grey arrows represent WMO recommendations for PWV retrieval.**

The transmittance $T(\lambda)$ of the whole atmosphere along the sun's direction can be expressed by the Bouguer-Lambert-Beer law (Swinehart, 1962):

$$T(\lambda) = \frac{I(\lambda)}{I_0(\lambda)} = e^{-m_r \tau_r(\lambda) - m_a \tau_a(\lambda) - m_g \tau_g(\lambda)} \ , \tag{1}$$

where $I(\lambda)$ is DNI recorded by the EKO instruments at wavelength $\lambda$, $I_0(\lambda)$ is the solar radiance at the top of the atmosphere, $m$ and $\tau$ is the air mass and optical thickness, respectively, the subscripts $r$, $a$ and $g$ denotes the contribution of Rayleigh, aerosols and other atmospheric gases, respectively (Bodhaine et al., 1999; Gueymard, 2001; Hansen and Travis, 1974). In the water vapor absorption band near 940 nm and 1370 nm, the absorption of other gases except water vapor can be neglected, the subscription $g$ in the above equation is replaced by $w$ , which means water vapor, and Eq. (1) can be rewritten as:

$$\frac{I(\lambda)}{I_0(\lambda)} = e^{-m_r \tau_r(\lambda) - m_a \tau_a(\lambda)} T_w(\lambda) \ , \tag{2}$$

$$T_w(\lambda) = \frac{I(\lambda)}{I_0(\lambda) e^{-m_r \tau_r(\lambda) - m_a \tau_a(\lambda)}} = \frac{I(\lambda)}{I_1(\lambda)} \ , \tag{3}$$

where $T_w$ is the transmittance within the water vapor band, $I_1(\lambda)$ is the radiance without water vapor absorption:

$$I_1(\lambda) = I_0(\lambda) e^{-m_r \tau_r(\lambda) - m_a \tau_a(\lambda)} \ , \tag{4}$$

In theory, water vapor correction on the spectral curve can fill in the water vapor absorption valley in the measured spectrum. Therefore, the radiance after removing the water vapor absorption $I_1(\lambda)$ can be approximated by interpolating the baseline points outside of the water vapor band. As shown by the dashed line in Fig. 4, besides the frequently used water vapor

absorption band near 940 nm, we also consider using the band near 1370 nm to invert the water vapor content in the dry atmospheres. The average water vapor transmittance within the water vapor band between $\lambda_1$ and $\lambda_2$ can be expressed as:

$$T_{w,\Delta\lambda} = \frac{1}{\Delta\lambda} \int_{\lambda_1}^{\lambda_2} \frac{I(\lambda)}{I_1(\lambda)} d\lambda ,  \tag{5}$$

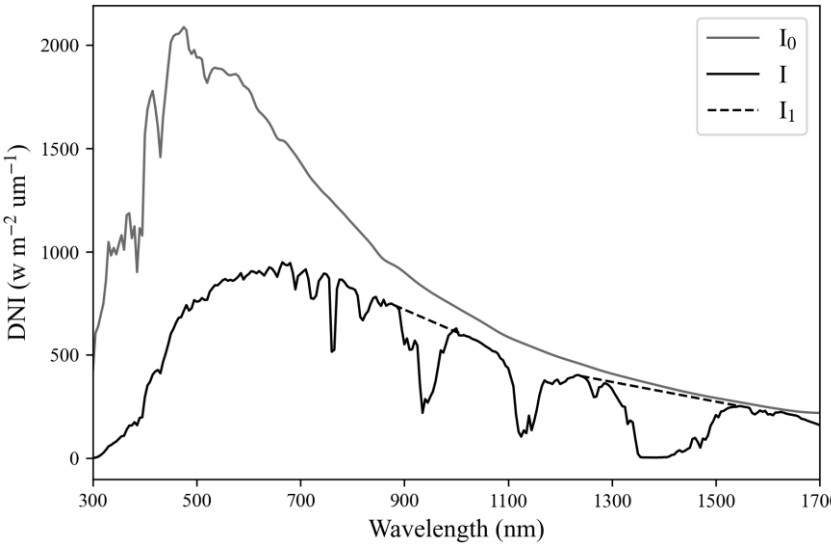


**Figure 4. Direct normal solar irradiance reaching the surface ($I$), the solar irradiance at the top of the atmosphere ($I_0$), and the irradiance after approximately removing the water vapor absorption by interpolating the baseline points outside the water vapor band ($I_1$).**

$T_{w,\Delta\lambda}$ can be given either by EKO spectroradiometers MS711 and MS712 denoted as $T_w^E$, or by radiative transfer model

(MODTRAN version 4.3), denoted as $T_w^M$ . In the model calculations, ignoring aerosol, cloud, and other gas absorption, the input atmospheric profile is the 1976 US Standard Atmosphere, and the FWHM is set approximately equal to the EKO instruments. The specific input parameters used in the calculations are listed in Table 2.

**Table 2 The input parameters to the MODTRAN model used in this work.**

| Parameters | Input parameters | References |
| --- | --- | --- |
| Boundary Aerosol Model | No aerosol or cloud attenuation | —— |
| Atmosphere profile | US Standard Atmosphere | NOAA (1976) |
| Altitude of surface | 0.05 km | —— |
| Slit function | Gaussian function, with FWHM of 6.5 nm | —— |
| Solar flux | 0.1 nm resolution | Kurucz (1994) |


$T_w^M$ was simulated with first guess of PWV and then the differences between $T_w^M$ and $T_w^E$ was calculated:

$$\Delta = T_w^E - T_w^M ,  \tag{6}$$

Recalculating Eq. (6) by increasing or decreasing PWV depending on that Δ is positive or negative, the final value of PWV is given by iteration of Eq. (6) as Δ becomes smaller than a criteria value:

$$\Delta \rightarrow min\left(\left|T_{w,\Delta\lambda}^{E} - T_{w,\Delta\lambda}^{M}\right|\right) \Longrightarrow PWV \,, \tag{7}$$

The PWV retrieval efficiency of BAND1 (900-990 nm) and BAND2 (1350-1450 nm) was tested separately using 1000 test spectral curves generated by dint of MODTRAN simulations. In the model simulations, the 1976 US standard atmosphere was used with random PWV between 0 and 0.35 cm, and solar zenith angle between 0°-30°, regardless of cloud and aerosol. Then the simulated spectral curves were superimposed with random noise within ±5 % at each wavelength to generate the test

spectral curves. Figure 5 shows the results of the inversion test of the two bands, the PWV retrievals of the band near 1370 nm are closer to the input PWV when the spectrum is simulated, and it is more stable, which demonstrates that the band around 1370 nm may be more suitable for water vapor retrieval in a dry atmosphere than the band around 940 nm.

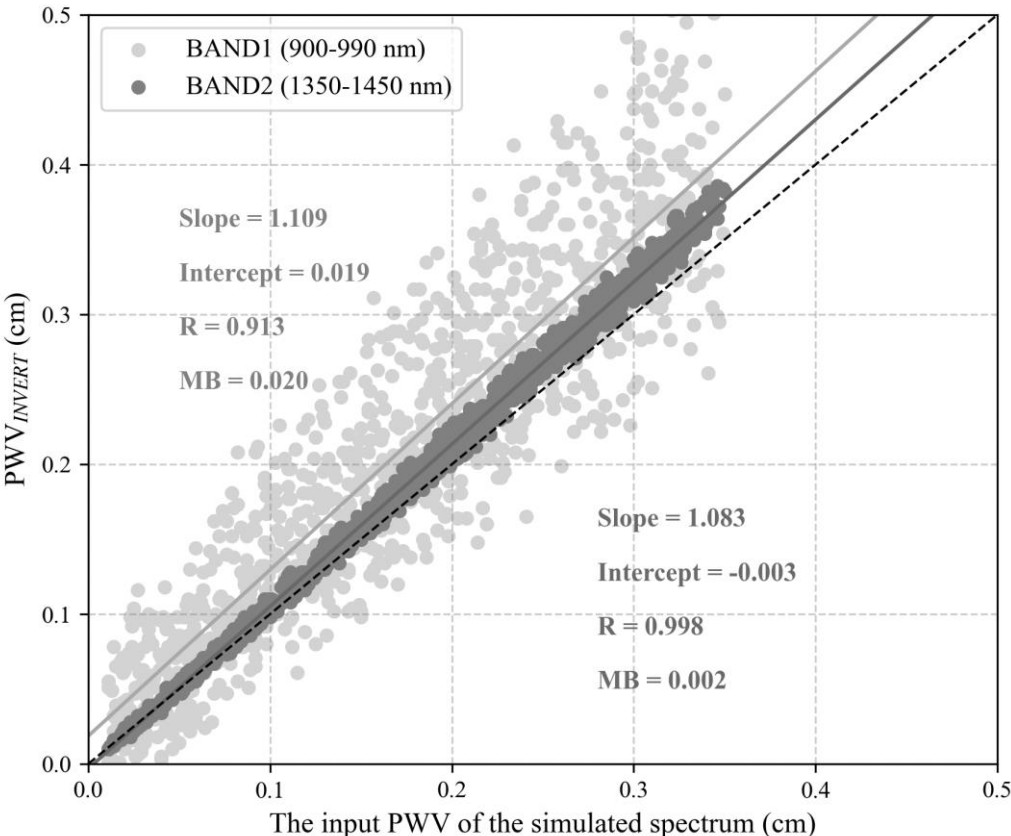

**Figure 5. Scatter plot of the PWV retrievals obtained from BAND1 and BAND2 of the test spectrum versus the input PWV of the**
**simulated spectrum and their linear fits.**

### 3.3 AOD inversion

After PWV is given, the spectral variation of AOD is derived according to Bouguer-Lambert-Beer law:

$$AOD = ln(I_0(\lambda)) - ln(I(\lambda)) - \tau_r - \tau_g , \tag{8}$$

$$\tau_r = p/p_0 \times 0.0088\lambda^{-4.05} , \tag{9}$$

$\quad \tau_g = \tau_{H_2O} + \tau_{N_2O} + \tau_{O_2} + \tau_{O_3} + \cdots , \tag{10}$

To mitigate the absorption of gases other than water vapor, the wavelengths used for AOD inversion are carefully selected, only the wavelengths at which the transmittance without the contribution of aerosol and Rayleigh scattering, greater than 0.999 are used. The AOD of other wavelengths was obtained by high-order fitting, specifically, as shown in Fig. 6. The Rayleigh

$\quad$ scattering $\tau_r$ is given by Eq. (9) (Ramachandran et al., 1994), $p_0$=1013.25 hPa, $p$ is provided by meteorological observation located in IAP, $\tau_{H_2O}$ is obtained from PWV inversion as described in Sect. 3.2.

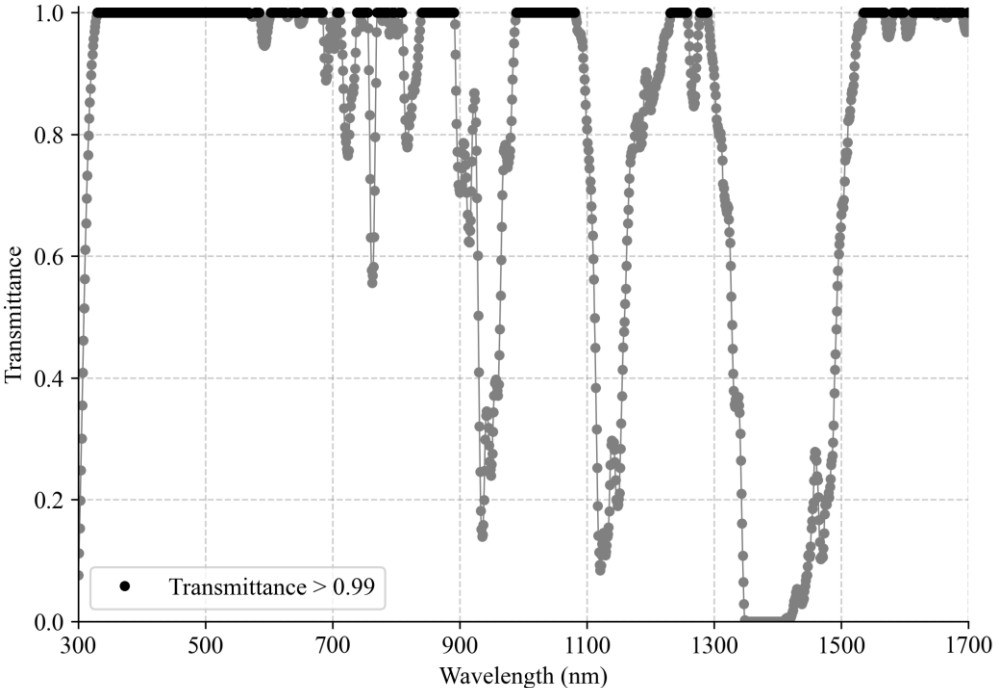

**Figure 6. The transmittance without Rayleigh scattering and continuous water vapor absorption in the EKO band, where the transmittance values greater than 0.99 are marked in black, and the rest are marked in grey.**

$\quad$ Since the FOV of the EKO spectroradiometers used in this work is 5°, besides the attenuated direct solar radiation, the scattered light from around the solar disk is also measured, and the recorded DNI is larger than the actual DNI, this may result in smaller AOD retrievals than the true aerosol optical depth (Sinyuk et al., 2012). Therefore, it is necessary to correct the CSR. As discussed by Blanc et al. (2014), the total radiation received by the instrument can be expressed as:

$$I' = \int_0^{2\pi} \int_{\alpha_0}^{\alpha_1} P(\xi) \, L(\xi) \cos(\xi) \sin(\xi) d\xi d\varphi , \tag{11}$$

Where $\xi$ is the half field angle, $\varphi$ is the azimuth angle, $P(\xi)$ is often called "penumbra function", which is 1 within the range of $\xi$ integration and 0 beyond the range, and $L(\xi)$ is the sky radiation, and it is simulated by the DISORT radiative transfer model (Stamnes et al., 1988a, b), as for EKO instruments, we set $\alpha_0 = 0.6°$ and $\alpha_1 = 2.5°$, so as to be consistent with the FOV of CE-318 sun photometer. The DNI after CSR correction can be expressed as:

$$I' = I - CSR, \tag{12}$$

The CSR Ratio (CR) in receiving direct radiation is expressed as:

$$CR = \frac{CSR}{I'+CSR}, \tag{13}$$

The aerosol optical depth after CSR correction can be expressed as:

$$AOD' = ln\big(I_0(\lambda)\big) - ln\big(I'(\lambda)\big) - \tau_r - \tau_g, \tag{14}$$

Combining equations (8), (13) and (14), the relationship between AOD and CR can be obtained:

$$AOD' = AOD + ln\left(\frac{1}{1-CR}\right), \tag{15}$$

Figure 7 shows the variation of CR with AOD at different wavelengths in the range of FOV from 1.2° to 5°. The aerosol data used in the simulation comes from the MERRA2 aerosol data, and the SZA is set to 30°. It shows a relatively larger difference due to the contribution of circumsolar radiation, especially for shorter wavelengths in an aerosol-laden environment.

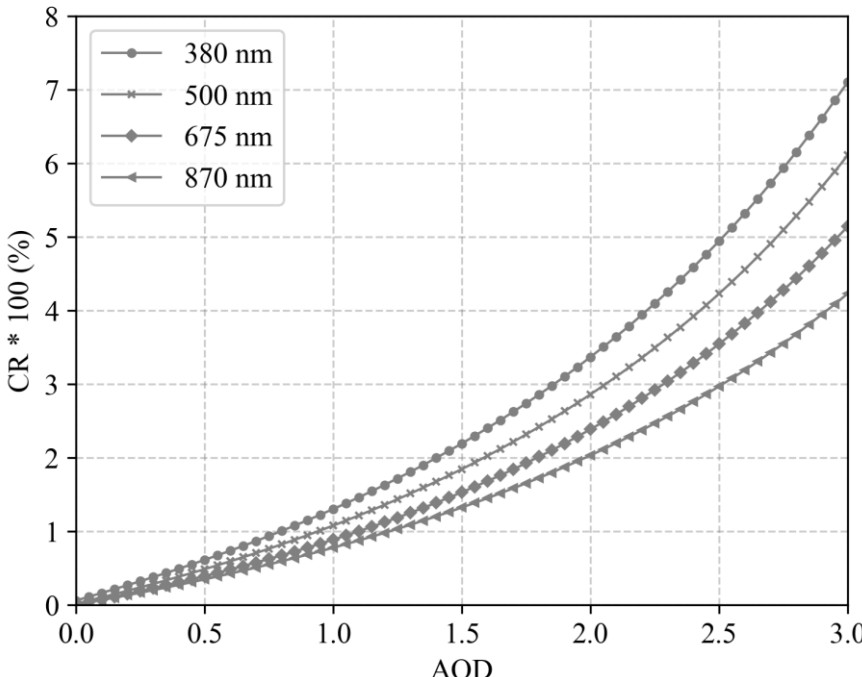

**Figure 7. Simulations of CR * 100 (%) for SZA 30∘ with AOD values from 0 to 3, at 380 nm, 500 nm, 675 nm, and 870 nm, for 2020 annual average MERRA2 aerosols data in the Beijing area with FOV between 1.2° and 5°.**

## 4 Uncertainty estimation of PWV and AOD retrievals

From the inversion method described in Sect. 3, it can be seen that the uncertainty of the inversion is mainly due to the spectral measurements of the EKO instruments and the retrieval algorithm. To estimate the uncertainty of the retrievals, 1000 spectrums

were generated by randomly superimposing the calibration uncertainty (Table 3) at each wavelength of two spectral curves (measured by EKO at 12:01 pm on 18 June 2020 and 12:10 pm on 13 December 2020), respectively. Afterward, PWV and AOD were inverted from these spectrums using the method described in Section 3, taking the standard deviation of the inversion values as the uncertainty of the inversions.

**Table 3 MS711 and MS712 calibration uncertainty**

| Spectroradiometer | Wavelength range | Uncertainty |
|---|---|---|
| MS711 | 300 nm – 350 nm | ±17.4 % |
| | 350 nm – 450 nm | ±5.1 % |
| | 450 nm – 1050 nm | ±4.2 % |
| | 1050 nm – 1100 nm | ±5.3 % |
| MS712 | 900 nm – 950 nm | ±4.52 % |
| | 950 nm – 1600 nm | ±4.84 % |
| | 1600 nm – 1700 nm | ±23.67 % |


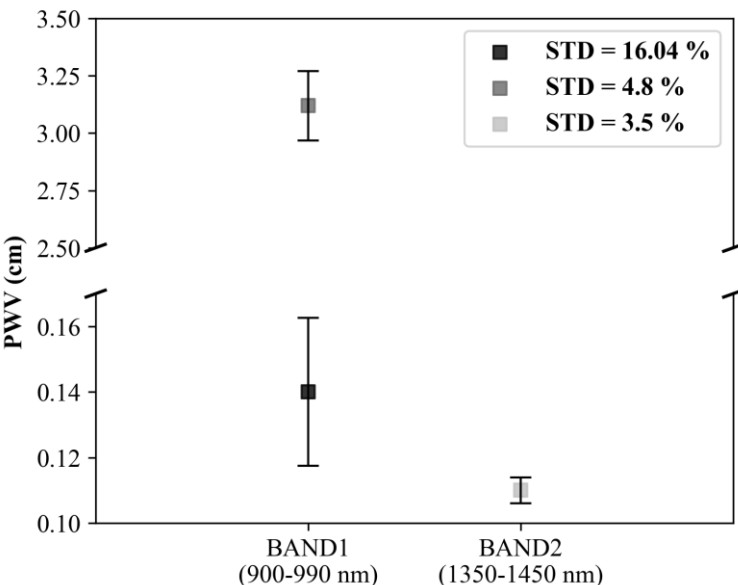

Figure 8. Mean and error bars of the PWV retrievals obtained using BAND1 and BAND2 based on the method described in Sect. 3.2 for the spectral curves after overlaying the calibration uncertainties.

Figure 8 shows the mean and error bars of the PWV retrievals using BAND1 and BAND2. The uncertainty of BAND1 inversions is 4.8 % at high water vapor content and 16.04 % at low water vapor content, and the uncertainty of BAND2 inversions at low water vapor content is 3.5 %. As can be seen, in the case of low water vapor content, the uncertainty of the PWV inversion of BAND1 is significantly larger than that of the rich water vapor content, but the uncertainty of the PWV inversion of BAND2 is still lower.

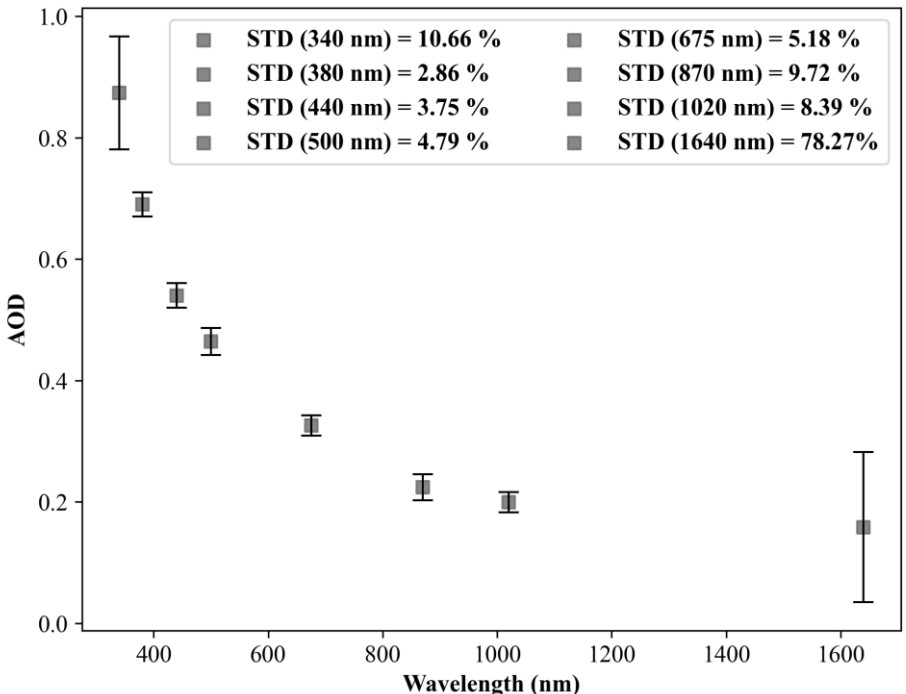


**Figure 9. Mean and error bars of AOD at the wavelengths corresponding to the CE-318 filters were obtained using the method described in Sect. 3.3 for the spectral curves after overlaying the calibration uncertainties.**

Figure 9 plots the uncertainties of the AOD retrievals at the wavelengths corresponding to the CE-318 filters. In general, the
uncertainties of AOD retrievals are low in the visible bands and increase in the near-infrared bands. Additionally, the larger uncertainties of the retrieved AOD at 340nm are due to unknown ozone amount and strong Rayleigh scattering, as for the larger uncertainty at 1640nm as shown in Fig. 9, may be due to the weak signals at this wavelength. Also, as pointed out by the EKO manufacturer, the calibration uncertainties of the EKO instruments at these two wavelengths are relatively large, and currently, AOD retrievals from the two wavelengths are not recommended by the author.

## 5 Results

The measurements of MS711 and MS712 from June 2020 to March 2021 at the top of IAP's building are used to derive the PWV and AOD, the space-time synchronized CE-318 data are used as the reference, and the number of matching data points is 5008. The mean deviation and variance between the results of the two instruments are given by:

$$\overline{X} = \frac{1}{n}\Sigma_i\left(X^i_{EKO}-X^i_{cimel}\right),$$ (16)

$$\delta_X = \sqrt{\frac{1}{n}\Sigma_i\left(X^i_{EKO}-X^i_{cimel}\right)^2},$$ (17)

where X is either PWV or AOD, the subscript denotes EKO instruments or CE-318.

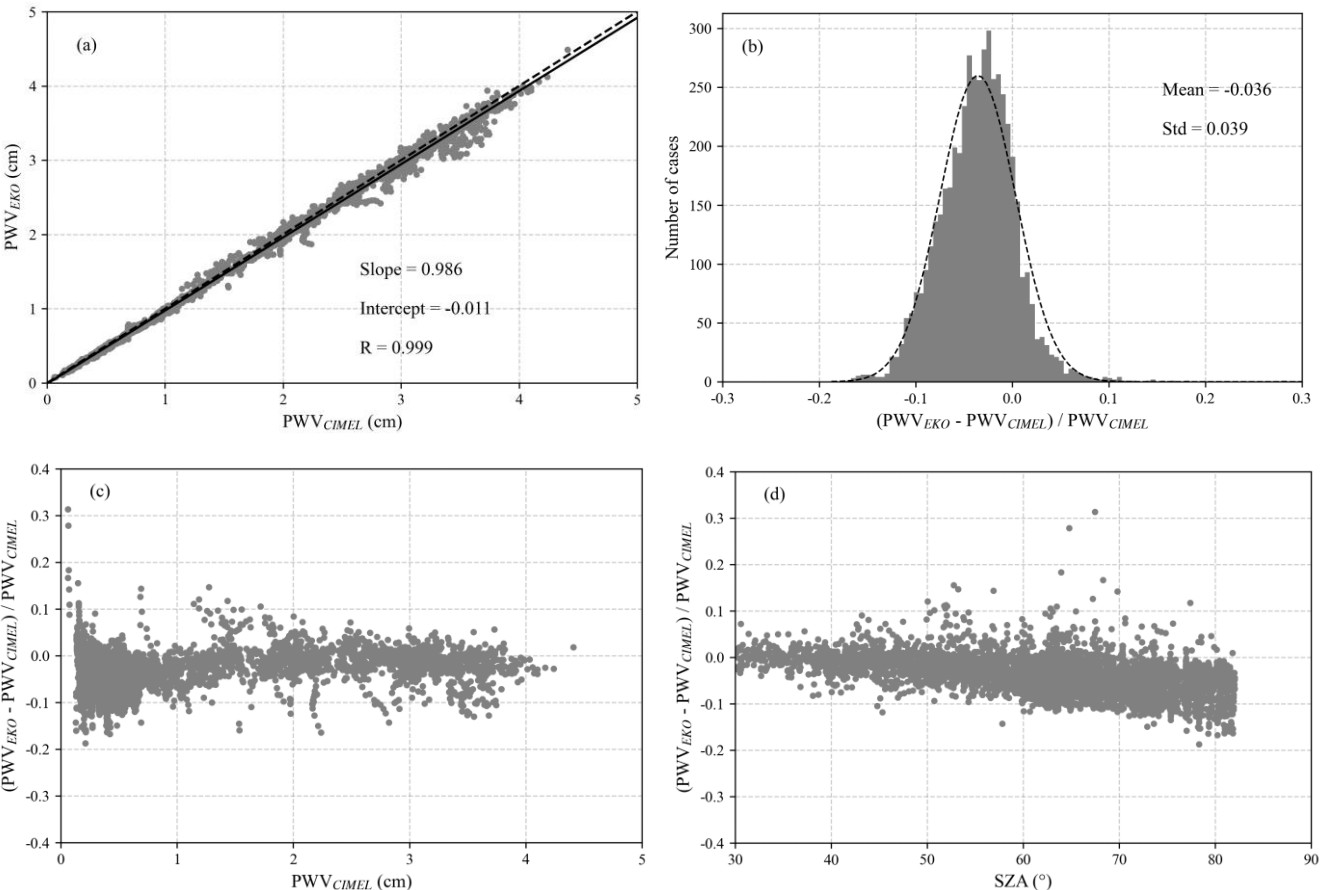

**Figure 10. PWV retrievals from EKO using the spectral approach in the 880–1000 nm region compared to the synchronous data of CE-318 in the measuring period (a), histogram of relative difference among PWV$_{EKO}$ and PWV$_{CIMEL}$ (b), and the relative difference plotted against PWV$_{CIMEL}$ (c) and solar zenith angle (d).**

The PWV retrievals using the band near 940 nm of EKO and CE-318 are shown in Fig. 10. It reveals that the retrievals of
EKO have a high consistency with those of CE-318, the correlation coefficient is 0.999, the mean bias and the standard
deviation are of -0.027 cm (-2.42 %) and 0.054 cm (3.93 %), respectively, the relative differences for 95 % of the retrievals
are between -0.114 and 0.042. Further analysis found that the differences are related to the solar elevation angle, the lower the
sun position, the larger difference. This is because in the case of a low solar elevation angle, the light intensity is very weak
and the light path is long, the uncertainty of the inversion will increase, resulting in a large deviation, which also occurs in
PWV inversion using other spectroradiometers (Kazadzis et al., 2014). In addition, as can be seen from Table 4, the relative
deviations of the PWV obtained by BAND1 (around 940 nm) varied from -2.04 % to -5.22 % for low water vapor content
(PWV < 0.5 cm), which is due to the increased uncertainty in PWV retrievals of the dry atmospheres.

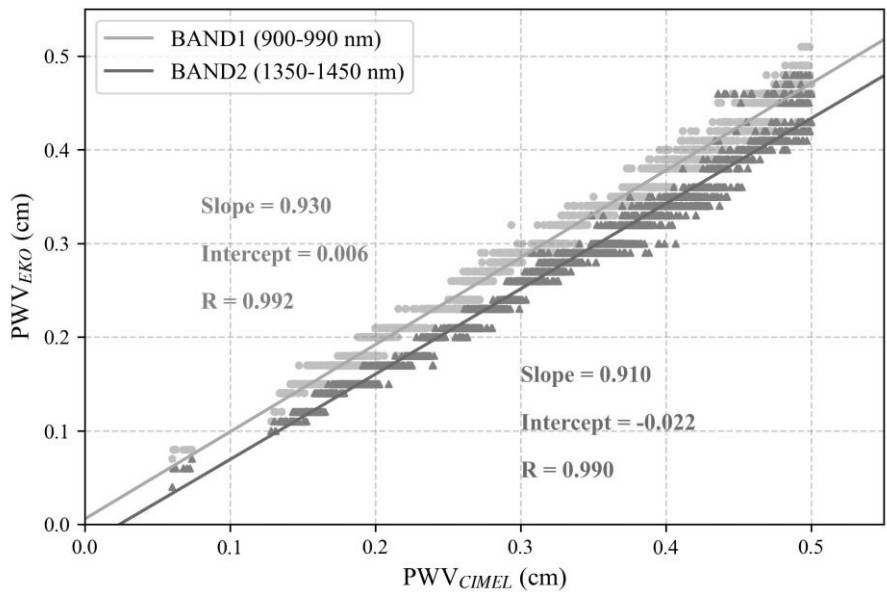

**Figure 11. Comparison of PWV retrieved from BAND1 and BAND2 with PWV$_{CIMEL}$ for PWV$_{CIMEL}$ less than 0.5 cm.**

**Table 4 Statistics of the comparison between PWV$_{EKO}$ and the PWV$_{CIMEL}$. (N: number of data, R: Pearson correlation coefficient, Slope: slope of the least squares fit between PWV$_{EKO}$ and PWV$_{CIMEL}$, RMSE: root mean square error, MB: mean bias, STD: standard deviation).**

| CE-318/EKO | BAND | N | R | Slope | RMSE (cm) | MB (cm) | STD (cm) |
|---|---|---|---|---|---|---|---|
| All data | BAND1 | 5008 | 0.999 | 0.986 | 0.061 (5.31 %) | -0.027 (-2.42 %) | 0.054 (3.93 %) |
| *PWV$_{CIMEL}$*>0.5 cm | BAND1 | 2977 | 0.998 | 0.985 | 0.077 (4.41 %) | -0.034 (-2.04 %) | 0.069 (3.50 %) |
| *PWV$_{CIMEL}$*<0.5 cm | BAND1 | 2031 | 0.992 | 0.930 | 0.022 (6.41 %) | -0.017 (-5.22 %) | 0.014 (4.13 %) |
|  | BAND2 | 2031 | 0.990 | 0.910 | 0.054 (16.79 %) | -0.051 (-15.60 %) | 0.016 (4.17 %) |

Figure 11 shows the PWV retrievals of BAND1 and BAND2 for dry conditions, their statistics are also listed in Table 4. The
results of BAND1 are relatively higher than those of BAND2, which is consistent with the theoretical simulation shown in Fig.

5. Therefore, we propose that for the dry environments and measurements are available, we can try to introduce the strong water vapor band near 1370 nm for PWV.

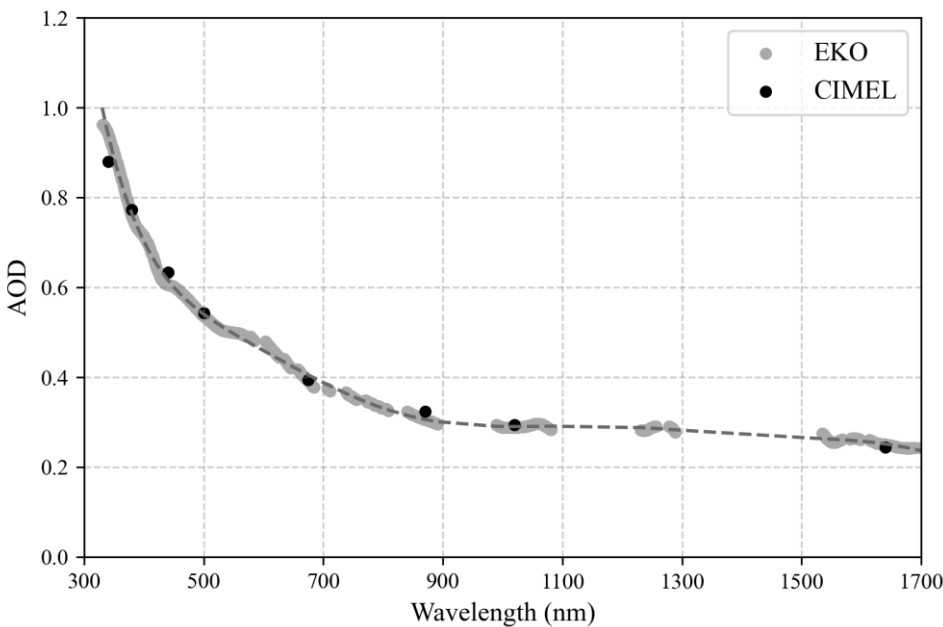

**Figure 12. The AOD was retrieved from EKO and CE-318 on 06 June 2020 (15:22 UTC+8), the dashed line is the spectral AOD obtained by the AOD$_{EKO}$ high-order fitting.**

Figure 12 shows an example of AOD$_{EKO}$, the AOD derived from EKO instruments is very close to the CE-318 data. The spectral AOD is obtained by the method described in Section 3.3. It is not suitable to provide spectral AOD in the case of ignoring the absorption of other gases except water vapor, so this example only illustrates that EKO instruments have the potential to provide spectral AOD.

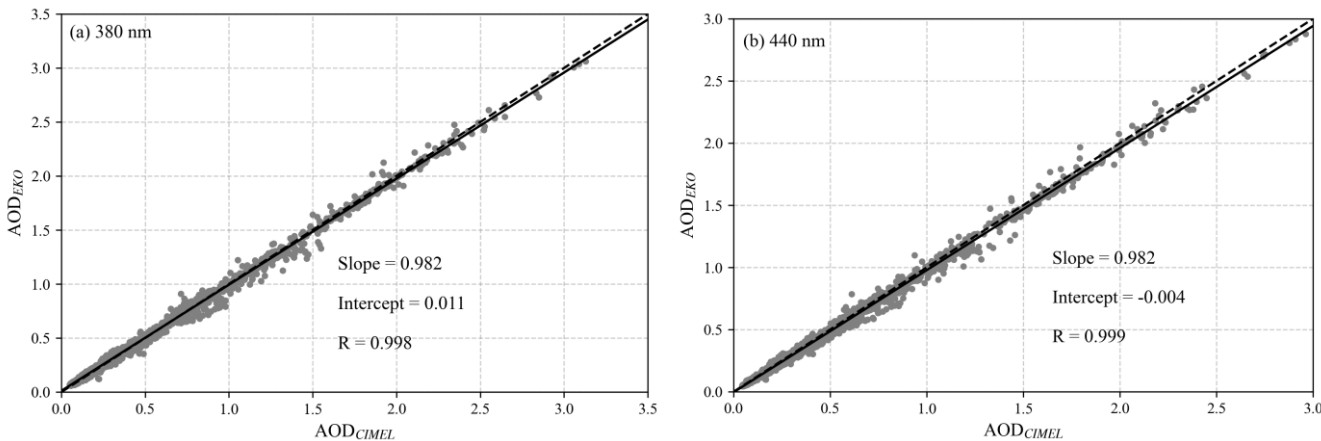

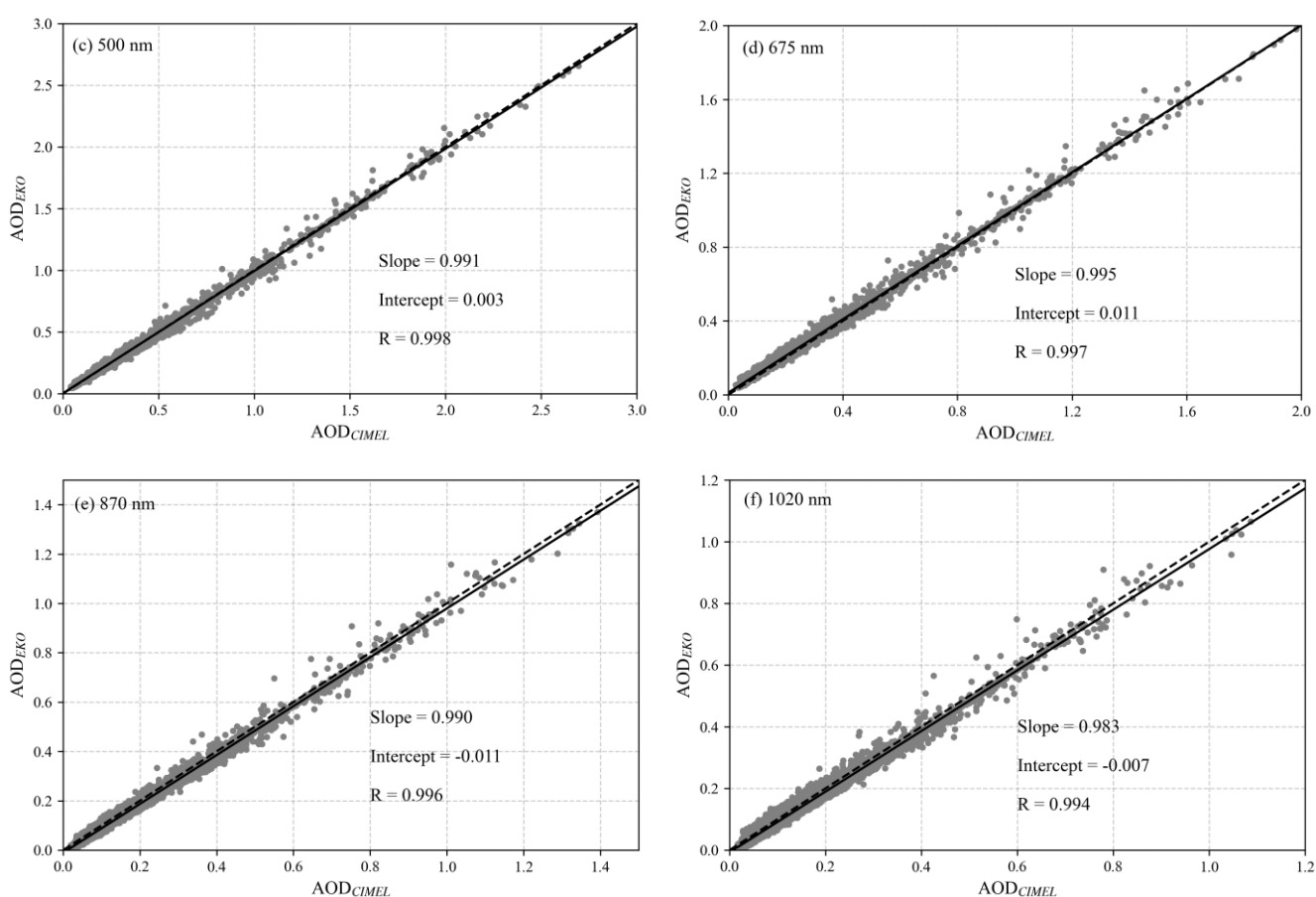

**Figure 13. Comparison of AOD$_{EKO}$ versus AOD$_{CIMEL}$ at 380 nm (a), 440 nm (b), 500 nm (c), 675 nm (d), 870 nm (e), and 1020 nm (f) from June 2020 to March 2021 at IAP.**

**Table 5 Statistics of the comparison between AOD$_{EKO}$ and AOD$_{CIMEL}$ at 380, 440, 500, 675, 870, and 1020 nm from June 2020 to March 2021 at IAP.**

| Wavelength (nm) | R | Slope | RMSE | MB | STD |
|---|---|---|---|---|---|
| 380 | 0.998 | 0.982 | 0.026 (9.98 %) | 0.003 (0.81 %) | 0.026 (8.89 %) |
| 440 | 0.999 | 0.982 | 0.024 (6.70 %) | -0.011 (-2.94 %) | 0.022 (5.70 %) |
| 500 | 0.998 | 0.991 | 0.020 (7.72 %) | <0.001 (-0.01 %) | 0.020 (7.54 %) |
| 675 | 0.997 | 0.995 | 0.021 (17.92 %) | 0.009 (4.27 %) | 0.019 (14.79 %) |
| 870 | 0.996 | 0.990 | 0.020 (19.87 %) | -0.012 (-7.31 %) | 0.015 (14.89 %) |
| 1020 | 0.994 | 0.983 | 0.018 (22.00 %) | -0.009 (-6.59 %) | 0.016 (18.82 %) |

To evaluate the differences between AOD$_{EKO}$ and AOD$_{CIMEL}$ ulteriorly, the AOD$_{EKO}$ in the corresponding bands of the CE-318 (380, 440, 500, 675, 870, 1020 nm) were compared and analyzed (Fig. 13). The specific statistics are listed in Table 5. The AOD retrievals from the two kinds of instruments are consistent, the correlation coefficients exceed 0.99, and the relative

differences are between -6.59 % and 4.27 %. Further analysis found that the AOD differences in the visible bands were small, especially at 500 nm, the MB and RMSE were <0.001 (-0.01 %) and 0.020 (7.72 %), respectively, while the differences in the near-infrared bands were significantly increased. According to the uncertainty analysis of AOD inversion in Sect. 4, it is probably because the uncertainties of AOD inversion are small in the visible bands but relatively large in the near-infrared bands.

## 6 Summary and Conclusions

The water vapor absorption band near 940 nm is currently used to derive the PWV commonly, and AOD from the sun photometer is usually given at several wavelengths apart, which sometimes does not fully meet the needs of the applications. Therefore, combined with the advantage of EKO instruments that can measure the direct normal solar irradiance in the spectral range of 300-1700 nm, the water vapor band near 1370 nm is also used to derive PWV for the dry atmospheres, and the spectral AOD is obtained by higher-order fitting of the AOD inverted from EKO at more wavelengths. Different from the three-parameter method, the retrieval algorithm is a physical method based on the radiative transfer model. Data measured by EKO MS711 and MS712 at IAP from June 2020 to March 2021 are used for inverting PWV and spectral AOD, and the results are compared with those from the collocated CE-318 sun photometer.

We used the calibration uncertainties obtained from the instruments calibration certificate to estimate the uncertainties of the water vapor and aerosol retrievals. The uncertainty of the PWV retrievals of the band around 940 nm at high water vapor content is significantly smaller than that at low water vapor content, ranging from 4.8 % to 16.04 %. The uncertainty of the PWV retrievals of the band near 1370 nm at low water vapor content is as low as 3.5 %. The uncertainties of AOD retrievals are large at wavelengths less than 350 nm and greater than 1600 nm, generally small in the visible bands (around 5 %), and relatively large in the other near-infrared bands (around 9 %).

The PWV retrieved from EKO instruments and CE-318 at the band near 940 nm are in good agreement, the correlation coefficient is 0.999, the mean bias, root mean square error, and standard deviation are -0.027 cm (-3.57 %), 0.061 cm (5.31 %), and 0.054 cm (3.93%), respectively. However, under dry conditions, there is little difference between the retrieved PWV from BAND1 (around 940 nm) and BAND2 (around 1370 nm), simulations through radiative transfer modelling show that the retrieved PWV with band 1370 nm is closer to the "true" value. Therefore, we proposed that a stronger water vapor band near 1370nm can be introduced for PWV retrieval at the dry atmospheres in case of the measurements are available.

The large FOV of the EKO instruments introduce more CSR into the measured DNI, which results in an underestimated AOD, and it must be corrected to approximate "true" AOD, especially for shorter wavelength under high aerosol loading. The AOD retrieved from EKO instruments after CSR correction agrees well with that from CE-318, the correlation coefficients are greater than 0.99, and the mean bias is between -0.012 and 0.009.

## Data availability

Data used in this study are available from the corresponding author upon request (dmz@mail.iap.ac.cn).

## Author contributions

M. Duan and C. Qiao determined the main goal of this study. C. Qiao carried it out, analyzed the data, and prepared the paper with contributions from all co-authors. S. Jia provided instrumental support. P. Wang and J. Huo provided guidance on algorithmic procedures.

## Competing interests

The authors declare that they have no conflict of interest.

**Acknowledgements**

This research is supported by the National Natural Science Foundation of China (Grant No. 42030107 and No. 42175150). We also thank all the teachers and students who participated in the discussion about this work.

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
