# Peer review of "Retrievals of Precipitable Water Vapor and Aerosol Optical Depth from direct sun measurements with EKO MS711 and MS712 Spectroradiometers"

_EGUsphere, 2022_

## Author Comment (AC1)

**Response to Referee #3:**

The authors greatly appreciate the helpful comments of the two referees. In the following, we present our point-by-point responses to the Referee #3. The referees' comments are in blue italic and our responses are in black. We have made appropriate changes in the revised manuscript by taking the comments into account.

**Overall evaluation:**

*The work of Qiao et al., focuses on the PWV and AOD retrievals of EKO MS711 and MS712. The method presented is based on methods of other instruments, but it is novel and since the spectral measuring instruments are becoming more popular, will become valuable in the future and fits the scope of AMT. However, the manuscript is poorly written, a lot crucial information for the reproducibility of the methodology are missing and the validation of the retrievals is very shallow. Hence, I suggest to be considered for publication after major changes.*

**Responses:** We greatly appreciate your valuable comments on our submitted manuscript. According to your comments, we have carefully revised the manuscript. The item-to-item responses to your comments are as follows.

*Specific comments*

*1. The two instruments are considered as one for most of the manuscript. I think it should be separated and make clear what is the performance of each one. Since the area around 940nm is overlapped by both them, the comparison of the measurements should be presented. Also, the different spectral steps and FWHM will result to very statistics in the validation process. It is crucial to present that, since the instruments are usually sold and installed separately and also in case of parallel operation, a decision should be made for the overlapping region. Finally, in section 2 more details should be mentioned such as the calibration of the instruments, the reported uncertainty and their measuring schedules. Specially, the calibration of the spectral bands is very*

*important and could lead to high deviations for the algorithm. Is there any wavelength shift? How are the spectral channels characterized?*

**Response:** Thank you for your comment. We reconfirmed that the radiometric measurements used in the 900-990 nm band are from MS711 recorded data. We found the relevant descriptions of the FWHM and wavelength accuracy of MS711 and MS712 from EKO Instruments official website. As can be seen from Figure 1(a) and (b), the full width at half maximum of both is <7nm, and the wavelength accuracy of both is ±0.2nm, so the wavelength drift hardly affects the inversion results. In addition, we contacted EKO in Japan and obtained the calibration certificates of MS711 and MS712. From the certificates, we learned that both instruments were accurately calibrated in Japan in 2018. The uncertainty of calibration is shown in Table 1. we performed a specific analysis of the inversion uncertainty due to the calibration uncertainty in Sect. 4 of the submitted manuscript.

[Figure]

[Figure]

**Figure 1 FWHM (a) and wavelength accuracy (b) of MS711 and MS712 (https://www.eko-instruments.com/us/categories/products/spectroradiometers/ms-711-spectroradiometer)**

**Table 1 MS711 and MS712 calibration uncertainty**

| Spectroradiometer | Wavelength range | Uncertainty |
|---|---|---|
| MS711 | 300nm – 350nm | ±17.4% |
| | 350nm – 450nm | ±5.1% |
| | 450nm – 1050nm | ±4.2% |
| | 1050nm – 1100nm | ±5.3% |
| MS712 | 900nm – 950nm | ±4.52% |
| | 950nm – 1600nm | ±4.84% |
| | 1600nm – 1700nm | ±23.67% |

*2. L75-80 More details should be provided on the cloud screening procedure. How effective was it? Give a figure showing the cloud screen data and discuss the results.*

**Response:** Thank you for your comment. We have added details related to cloud screening in Sect. 3.1 of the manuscript, drawn the graphs of cloud detection instances, and discussed the detection effect.

*3. It is implied that the radiative transfer model used is MODTRAN. Please, add a section in 2, about the model, the setup, the selection of variables and the bibliographical accuracy.*

**Response:** Thank you for your comment. We added the detailed parameter settings of the mode and the corresponding bibliography as the table 2 in the Sect. 3.2. (Lines 145-149)

**Table 2 The input parameters to the MODTRAN mode used in this work.**

| Parameters | Input | Reference |
|---|---|---|
| Boundary Aerosol Model | No aerosol or cloud attenuation | —— |
| Atmosphere profile | US Standard Atmosphere | NOAA. (1976) |
| Altitude | 0.05 km | —— |
| Slit function | Gaussain function, with FWHM of 6.5 nm | —— |
| Radiative transfer | DISORT | Stamnes et al. (1988) |
| Solar flux | Kurucz (0.1 nm resolution) | Kurucz (1994) |

*4. In general the 1370 absorbing window is more sensitive to PWV changes, but the Direct Irradiance signal at this spectral range is a lot lower. Hence, before using it, signal to noise ratio for the instrument should be discussed and the expected uncertainty should be estimated.*

**Response:** Thank you for your comment. We use MODTRAN mode to simulate and find that when the PWV is greater than 3cm(SZA=0°), the water vapor transmittance at 1370nm is zero, so theoretically the measurement value of the instrument at 1370nm should also be zero, if not, it is considered to be the measurement noise of the instrument at 1370nm. Therefore, we use the mean value of the measurements at 1370nm corresponding to the PWV inversion value greater than 3.5cm as the background noise signal and estimate that the SNR of the band near 1370nm used for inversion of PWV in a dry environment is greater than 60. In addition, uncertainty analysis for water vapor inversion in the band around 1370 nm is added in Sect. 4 of the manuscript.

*5. L103/figure 4. This approach should be discussed thoroughly and the results need to be evaluated.*

**Response:** Thank you for your comment. We have added a description of the method in the manuscript (lines 131-153), and then the water vapor inversion efficiencies of the two water vapor absorption bands were compared and analyzed. ( lines 154 -164)

*6. L117 Results showed in figure 5 are not enough to prove that one band is more efficient than the other. We don't know what is the testing sample, how representative is and all other effects on the measurements are already eliminated. A discussion leading to figure 5 is clearly missing.*

**Response:** Thank you for your comment. I am very sorry that our explanation of this part is not clear enough in the previous manuscript. In order to test the PWV retrieval efficiency of BAND1 and BAND2, the spectral curves used in our test are based on MODTRAN simulations with random noise added afterwards. In spectral simulations, the US standard atmospheric model was used with random PWV between 0-0.5 cm, and solar zenith angle between 0°-30°, regardless of cloud and aerosol. Then the simulated spectral was superimposed a random noise of $\pm 5$ % on each wavelength to generate the test spectral curves. Figure 2 shows the results of the inversion test using the two bands, the PWV retrievals of the band near 1370nm are closer to the input PWV when the spectrum is simulated, and it is more stable, which demonstrates that the band around 1370 nm may be more suitable for water vapor retrieval in dry atmosphere than the band around 940 nm for the water vapor inversion using the method in Sect. 3.2.

The result plot of the inversion test is different from Figure 5 in the previous manuscript because noise was previously added to the overall spectrum of the MODTRAN output. When checking the manuscript, considering that this was not reasonable, we redid the inversion test and added noise on each wavelength of the mode output spectrum.

[Figure]

**Figure 2. Scatter plot of the water vapor retrievals obtained from BAND1 and BAND2 of the test spectrum versus the input PWV of the simulated spectrum and their linear fits.**

*7. 3.2 It is not clear at which wavelengths this inversion will be used. It is a odd to name this aod inversion in general, since it is not valid for the most wavelengths (where other gases absorb). I suggest to focus in water vapor bands and close bandwidths and just calculate aod for those and keep the full aod inversion for future work that will include more trace gases.*

**Response:** Thank you for your comment. We have added Figure 3 to Section 3.3 of the manuscript, marking the wavelengths used for AOD inversion, specifically those with transmittance greater than 0.999 excluding Rayleigh scattering and continuous absorption by water vapor. The AOD for other wavelengths are obtained by high-order fitting. In addition, thanks for your suggestion, we have stated in the article that the EKO instruments have the potential to provide spectral AOD, and we are also considering your suggestion to use the spectral AOD of this instrument for ohter trace gases retrieval.

[Figure]

**Figure 3. The transmittance without Rayleigh scattering and continuous water vapor absorption in the EKO band simulated by MODTRAN, where the transmittance value greater than 0.999 is marked in black, and the rest are marked in grey.**

*8. L139 these uncertainties should be discussed and estimated in a separate section. Also, the fact that is compared with CIMEL retrievals, which was found in other studies to drift above 70° sza.*

**Response:** Thank you for your comment. We have added a section to the manuscript to estimate the uncertainties of water vapor and aerosol retrievals. Since we use the physical method for inversion, it is not convenient to use the conventional error propagation method to measure the uncertainties, therefore, we use another approach to estimate the uncertainties, which is described in Sect. 4 of the manuscript. In addition, some explanations are added for the increase of the difference between the EKO PWV retrievals and the CIMEL PWV retrievals when the zenith angle is greater than 70°. (Lines 223-226)

*9. "here we say", I don t understand this phrase.*

**Response:** Thank you for your comment. The phrase has been removed, and the article has been carefully checked for grammar and expression.

*10. Figure 7 discussion. It is clear that band 2 is underestimating PWV constantly. It is more like a constant bias of 0.02 between the two bands. So this seems more a calibration issue (between the model and the instrument) than a systematic error of the method.*

**Response:** Thank you for your comment. We also found such an underestimation. From the calibration certificate provided by EKO Instruments, we see that the calibration uncertainties for the two bands used for water vapor inversion are not very different, which are ±4.2% and ±4.84%, respectively. In addition, when using EKO data, the PWV retrievals of BAND2 is lower than that of BAND1, which is consistent with the result of the inversion test using the test spectrums, however, the added uncertainties on BAND1 and BAND2 of the test spectrums are the same. Therefore, we speculate that such underestimation is not caused by the calibration differences, and certainly cannot be completely excluded from differences in MS711 and MS712.

*11. Figure 2.L82 This figure does not show water vapor absorption windows. It is just two random measurements. Do we know that there was different PWV at these days? Figure 3 Clearly shows the windows, but the figure 2 has no use at this version of the manuscript.*

**Response:** Thank you for your comment. This figure has been deleted from the manuscript.

*12. Figure 3. I don't understand the purpose of visualizing cimel filters. Also, the aerosol line, corresponds to a specific AOD (which will change the transmittance). Please change the legend to the actual AOD value. Also, move the legend to a position that does not hide the drop at 1300-1500nm.*

**Response:** Thank you for your comment. Referring to your suggestion, we have changed the original Figure 3 to Figure 4. The spectral response curves of the CIMEL filters in the figure are drawn to show the position of the CIMEL measurement band more intuitively.

[Figure]

**Figure 4. The spectrum response curves of CE-318 photometer's filter wheels, and the transmittance of water vapor, aerosols and Rayleigh scattering in the spectral range of 300–1700 nm, which are calculated by MODTRAN4.3 at SZA=0°, PWV=0.5 cm, PWV=3.0 cm and Boundary Aerosol Model=Rural extinction(spring-summer), VIS=23 km. The wavelengths pointed by the grey arrows represent WMO recommendations for PWV retrieval.**

*13. Figure 4. Describe better at the caption. Information on how these spectra were retrieved.*

**Response:** Thank you for your comment. We have changed the original caption **"Figure 4. Direct normal solar irradiance reaching the surface, $I$, the irradiance after removing water vapor absorption, $I_1$, and the solar irradiance at the top of the atmosphere, $I_0$."** to **"Figure 4. Direct normal solar irradiance reaching the surface ($I$), the solar irradiance at the top of the atmosphere ($I_0$), and the irradiance after approximately removing the water vapor absorption by interpolating the baseline points outside the water vapor band ($I_1$)."**

*14. Figure 5.    What are the "real values"? If it is CIMEL retrievals, keep in mind that previous studies showed that CIMEL was the most erroneous from all the methods (GPS, radiosondes, microwave radiometer).*

**Response:** Thank you for your comment. Figure 5 in the original manuscript shows the results of examining the water vapor inversion efficiency of BAND1 and BAND2 using the test spectrums. Here "real values" refers to the water vapor input values when

simulating the spectrums. Considering the inappropriate use of the language, we made a change from "real values" to " The input PWV of the simulated spectrum".

*15. Figure 8. It is not wise to provide spectral AOD, when all the trace gases but the water vapor are ignored.*

**Response:** Thank you for your comment. It is indeed unreasonable to provide the AOD of the spectrum ignoring the absorption of gases other than water vapor, so we have added some clarifications to the manuscript in conjunction with your suggestion and changed Figure 8 in the previous manuscript to the following figure. (Lines 240-247)

[Figure]

**Figure 5. The AOD was retrieved by EKO and provided by AERONET-CIMEL on 06 June 2020 (15:22 UTC+8), the dashed line is the spectral AOD obtained by the AOD$_{EKO}$ high-order fitting.**

---

## Author Comment (AC2)

**Response to Referee #2:**

The authors greatly appreciate the helpful comments of the two referees. In the following, we present our point-by-point responses to the Referee #2. The referees' comments are in blue italic and our responses are in black. We have made appropriate changes in the revised manuscript by taking the comments into account.

**Overall evaluation:**

*The authors provide us a physical algorithm based on radiative transfer to derive the Precipitable Water Vapor (PWV) and Aerosols Optical Depth (AOD) from a pair of ground-based spectroradiometers, EKO MS711 and MS712. In their algorithm the water vapor band near water vapor band near 1370 nm as well as near 940 were also used to derive the PWV, this is very important for dry atmosphere, e.g. for very cold area such as Tibet or other high altitude plateau.*

*All my questions and concerns in my first review has been answered, and a new revised manuscript was submitted, I have no more concerns for this revised version except a few typo mistakes and language sentences. I agree it publishing in AMT after the revision.*

**Responses:** We greatly appreciate your valuable comments on our submitted manuscript. According to your comments, we have carefully revised the manuscript. The item-to-item responses to your comments are as follows.

*Minor concerns:*

*1. Line 31-33, "MWPS measures the radiation emitted from the atmosphere by microwaves, yields a vertical profile of water vapor, which can then be integrated to give PWV, where aerosols have little effect, but this measurement is very expensive (Güldner and Spänkuch, 2001; J. and Güldner, 2013)." should be rewritten as: "MWPS measures the radiation emitted from the atmosphere by microwaves, yields a*

*vertical profile of water vapor, which can then be integrated to give PWV (Güldner and Spänkuch, 2001; J. and Güldner, 2013). The advantages of using microwave for PWV is that aerosols have little effect, but the disadvantage is that this kind of instruments is generally is very expensive"*

**Response:** Thank you for your comment. We have corrected the corresponding expression in the article according to your suggestion. (Lines 33-34)

*2. Line 39, the short name "PHOTOS" should be clearly mentioned the first time.*

**Response:** Thank you for your comment. We are sorry that we did not check the previous manuscript carefully enough, we have corrected the mistakes in the article and rechecked the spelling of the manuscript again.

*3. Line 89, it's better to replace "based on this" as "therefore"*

**Response:** Thank you for your comment. we have made corresponding changes in line 112 of the article.

*4. Line 114-116 "When simulating spectral curves, the US standard atmospheric model was selected, regardless of clouds and aerosols, randomly inputted PWV of 0-0.5 cm and solar zenith angle of 10°-45°, and superimposed -1 %-+1 % noise on the simulated curves." Should be rewritten to make it clear, maybe better written as: "In spectral simulations, the US standard atmospheric model was used with random PWV between 0-0.5 cm, and solar zenith angle of 10°-45°, regardless of clouds and aerosols. The simulated spectral plus +/-1% was used for retrieval."*

**Response:** Thank you for your comment. We have corrected the corresponding expression in the article according to your suggestion. (Lines 155-158)

---

## Referee Report (RR1)

Review of Qiao et al. (submitted to AMT)

I find the quality of this work to be questionable, as will be revealed by several comments below. With this in mind, I don't really have a compelling reason to see this paper published. Had the authors succeeded in demonstrating the utility of the 1370 nm band with real data, I would be in favour of publication, but, as is, it could be that the lack of success is simply due to weaknesses in their method (see specific weaknesses below and also the neglect of forward scattering).

I agree with many of the comments raised previously by Reviewer 1 and find that Reviewer 2 was not thorough. Specifically, the point raised by Reviewer 1 about MS711 and MS712 being very different instruments is noteworthy. These instruments should be validated separately. Also, MODTRAN4.3 is used, which relies on HITRAN1996 (I believe). This is outdated. MODTRAN 5 and 6 are available and the spectroscopic parameters for water vapor have changed. I know that the Brown et al., 2002 parameters are used in MODTRAN 5.2 for example:

Brown, L. R., Toth, R. A., and Dulick, M.: Empirical line parameters of $H_2^{16}O$ near 0.94 μm: Positions, intensities and air broadening coefficients, J. Mol. Spectrosc., 212, 57–82, 2002.

It seems that the main point of Fig. 3 was to show the absorption due to water vapour at ~940 nm is small when PWV is 0.5 cm, but the figure shows that absorption is quite strong. The comparison of the suitability of the 940 and 1370 nm bands for PWV retrieval is one of the main points of the paper. The authors appear to hypothesize that the stronger 1370 nm band would be more suitable for a dry atmosphere, but their results with real data do not make a strong case for this longer wavelength band.

5% (relatively) uniform noise throughout the two bands is not realistic. A proper instrument model should be used since SNR will decrease at wavelengths for which the transmitted irradiance is low in the case of a grating spectrometer. This could be why the results are quite different between simulated data (Fig. 5) and real data (Fig. 10).

L17: "CE-318" -> "CE-318 sun photometer."

L17: Delete "which shows that" and start a new sentence with "The two …"

L30: I don't know if acronyms need to be defined in the main body of the paper to make it independent of the abstract.

L31 (and L41): "etc." should not be used like this. "and others" is preferable.

L41: I suggest deleting "etc." here.

L44: Define PHOTONS

L44: CE-318" -> "the CE-318"

L48: Delete "the"

L51: This sentence does not give the spatial domain. Is it a global comparison? 4 references are provided at the end of the sentence but it is not clear which one contains the standard deviation of PWV retrieval differences and the overestimation of AOD.

L52: End this long sentence after "CE-318".

L52: The second half of this sentence "and the PWV given…" does not make sense. What is meant by "integration"? 940 nm is mentioned, so what are the "different wavelengths"?

L52: "single" -> "a single"

L63: environment -> environments

L63: There should be a sentence stating whether and why 1370 nm would be less useful in humid environments.

L88: need -> needs

L100: Delete the two consecutive sentences starting with "The light grey"

L107: 0 °solar -> 0° solar

L113: atmosphere -> atmospheres

L146 (Table 2): mode -> model

L146 (Table 2): "Altitude" –> "Altitude of surface"

L146 (Table 2): DISORT is not needed for modelling transmittance. DISORT is used when scattered light into the field of view is considered.

L151: Give the update equation (i.e. how is the increase/decrease of PWV calculated from iteration to iteration). Is Chahine's method used?

L156: The PWV might be randomly generated but it the upper bound is clearly not 0.5 as shown in Fig. 5.

L156: "0-0.5" -> "0 and 0.5"

L156-157: Delete "and generating 1000 simulated spectral curves."

L160: "in" -> "in a"

L162 (Fig. 5 and Fig. 10): use grey font for the statistics for Band 1.

L168: The transmittance due to Rayleigh scattering is approximate. Better formulations exist, but this might be OK since 340 nm seems to be the shortest wavelength used for AOD.

L171: "the wavelengths used for AOD inversion were carefully selected by using MODTRAN to calculate and filter the wavelengths corresponding to the transmittances greater than 0.999 that do not include Rayleigh scattering and continuous water vapor absorption" -> "the wavelength used for AOD inversion are those for which the MODTRAN transmittances excluding Rayleigh scattering and water vapor absorption are greater than 0.999."

L175: "as" –> "as described in"

L177 (Fig. 6): 0.99 -> 0.999        (in legend)

L184: on -> from

L190 (Fig. 7 caption): Either here or preferably in Table 2, the authors should clearly state the average PWV for the "low" or "rich" water content. There is no mention of what 'rich' means quantitatively in this paper.

L204: "Presumably due to the large calibration uncertainty…" This explanation is rather simple. Also, the first part of this sentence really belongs at the end of the previous sentence. These two extreme wavelengths have large AOD relative uncertainties for non-instrumental reasons. At 340 nm, aerosols contribute more weakly to the total extinction because of strong Rayleigh scattering and ozone absorption. At 1640 nm, aerosol extinction is weak, particularly when particle size is small (radii < 160 nm). This brings me to another point. Table 2 says there is no boundary layer aerosol used. So how is AOD retrieved? Show the true AOD at each wavelength in Fig. 8.

L225: The discussion here needs to be improved. The authors' statement should be clear that biases do not result from low signal-to-noise ratio.

L228: "with decreasing" -> "for low"

L229 (Fig. 10): The 1370 nm band is worse in terms of slope, intercept, and R.

L235: "may be more accurate" is not supported by the evidence (Fig. 10). Speculative statements should be removed even if they are logically defensible.

L247: Why would AOD provide "some assistance" for retrieval of other trace gases from direct sun measurements?

L247: "other trace gases retrieval." -> "retrieval of abundances of other trace gases."

L258. Start a new sentence: "The specific …"

L293: Very questionable speculation…. should be removed unless it can be supported using real data.

L296: are -> is

L298: Start a new sentence after "wavelengths": "This will be considered in the future version."

---

## Referee Report (RR2)

Review of Qiao et al. (submitted to AMT)

I forgot that I had reviewed an earlier draft of this paper and reviewed this version "independently" of the previous one. I find the same issues I had with the previous version. I think the paper is quite good on the whole, but there are a couple of unsolved problems:

1) MODTRAN 4.3 is outdated.
2) Fig. 11 shows that the slope, intercept and correlation coefficient are all worse for the 1.36 micron band, so the statement on L297 is not supported. Section 6 contains a similar statement.

L37: "economical to build observation network" -> "it is economical to build an observation network with them"

L55: "940nm" -> "940 nm"

L61: Why is the "three-parameter formulation method" very sensitive to 'air quality'? I don't believe this is true since path lengths are geometric. Also "formulation method" is redundant in my opinion.

L65: "easily not" -> "not easily"

L78: 10-15 -> 3

L79: resolution is the wrong word. 'bandwidth' is better. Are these values true for both UV bands and for the three visible bands and for the 4 near-IR bands?

L86: are -> is

L90: Considering -> Consider

L91: are -> is

L130: 'completely' is an adverb and does not belong here.

L160: Note that random noise cannot give a biased slope, non-zero intercept or any MB. This is simply a point of information, no need to change the wording.

L169: great -> greater

L176: extenuated -> attenuated

L194: "under high aerosol loading atmosphere" -> "in an aerosol-laden environment"

L223: "above" -> "the above"

L255: atmosphere -> conditions

L257: atmosphere -> periods                    (or conditions)

L295: "at the" -> "at"

L296: "near-infrared band" -> "other near-infrared bands"

L299: "a" -> "a dry"

L300: This conclusion is not supported by the results. Please reword. Maybe consider limiting to even drier conditions.

---

## Author Response (AR2)

**Response to Referee #4:**

The authors greatly appreciate the helpful comments of referee #4. In the following, we present our point-by-point responses to the Referee #4. The referees' comments are in blue italic and our responses are in black. We have made appropriate changes in the revised manuscript by taking the comments into account.

**Main comments:**

*1. I find the quality of this work to be questionable, as will be revealed by several comments below. With this in mind, I don't really have a compelling reason to see this paper published. Had the authors succeeded in demonstrating the utility of the 1370 nm band with real data, I would be in favour of publication, but, as is, it could be that the lack of success is simply due to weaknesses in their method (see specific weaknesses below and also the neglect of forward scattering).*

**Responses:** Thank you for your comment. The PWV results obtained by using the water vapor absorption band near 1370 nm with the real data. Figure 5 and Figure 11 in this revised manuscript shows the same conclusion, that is, for a dry atmosphere, the PWV obtained by using the water vapor absorption band near 940 nm (BAND1) is higher than that of the band near 1370 nm (BAND2). It can be seen from Fig. 11 that the PWV retrievals of BAND1 are closer to those of CE-318 sun photometer, which demonstrated that the algorithm is OK, because it is not easy to say which PWV obtained by using BAND1 and BAND2 is closer to the 'true' value, Therefore, we used the model to simulate some theoretical spectra for the inversion test. In the test results (Fig. 5), the PWV retrievals obtained by using BAND1 are also higher than those of BAND2, which is consistent with Fig. 11, while the PWV retrievals from BAND2 are closer to the theoretical inputs, which are assumed to the "True" value. Therefore, we proposed that for dry environment, we can try to introduce the stronger water vapor band around 1370 nm for PWV inversion when measurements are available.

Thank you for pointing out the problem of the forward scattering issue due to the large FOV of the EKO instruments (5°), we did neglect the possible impact in previous algorithm. As shown in Fig. 1, the Circumsolar radiation Ratio (CR) at infrared wavelengths (870 nm) is very small, which will not affect the water vapor retrieval. However, in the inversion of aerosol optical depth, especially for high AOD loading atmosphere and at shorter wavelengths, it is sure necessary to perform CSR correction. We have added CSR correction in the revised manuscript.

[Figure]

**Figure 1. Simulations of CR * 100 (%) for SZA 30 ° with AOD from 0 to 2, at 380 nm, 500 nm, 675 nm and 870 nm, for 2020 annual average MERRA2 aerosols data in Beijing area for FOV between 1.2 ° and 5 °.**

*2. I agree with many of the comments raised previously by Reviewer 1 and find that Reviewer 2 was not thorough. Specifically, the point raised by Reviewer 1 about MS711 and MS712 being very different instruments is noteworthy. These instruments should be validated separately. Also, MODTRAN4.3 is used, which relies on HITRAN1996 (I believe). This is outdated. MODTRAN 5 and 6 are available and the*

*spectroscopic parameters for water vapor have changed. I know that the Brown et al.,*

*2002 parameters are used in MODTRAN 5.2 for example:*

*"Brown, L. R., Toth, R. A., and Dulick, M.: Empirical line parameters of $H2^{16}O$ near 0.94 μm:*

*Positions, intensities and air broadening coefficients, J. Mol. Spectrosc., 212, 57–82, 2002."*

**Responses:** Thank you for your comment. We have accepted the suggestions and comments proposed by reviewer 1. MS711 and MS712 have the same full width at half maximum (FWHM) and wavelength accuracy, and the possible impact of the differences between the two instruments has not been considered in the relevant applications of this manuscript. We noticed that the current version of MODTRAN4.3 may be old, so the water vapor transmittances were calculated under the same conditions using MODTRAN5.2 and MODTRAN4.3. Figure 2 shows the results of water vapor transmittances calculated using two versions of MODTRAN model. Because a narrow band other than single wavelength is used, the water vapor transmittances calculated by MODTRAN4.3 and MODTRAN5.2 are almost the same.

[Figure]

**Figure. 2 Water vapor transmittances calculated by MODTRAN4.3 and MODTRAN5.2 under the same conditions, respectively (SZA=0 °,PWV=1.0 g*m$^{-2}$,FWHM=7 nm).**

*3. It seems that the main point of Fig. 3 was to show the absorption due to water vapour at ~940 nm is small when PWV is 0.5 cm, but the figure shows that absorption is quite strong. The comparison of the suitability of the 940 and 1370 nm bands for PWV retrieval is one of the main points of the paper. The authors appear to hypothesize that the stronger 1370 nm band would be more suitable for a dry atmosphere, but their results with real data do not make a strong case for this longer wavelength band.*

**Responses:** Thank you for your comment. Figure 3 in the manuscript shows that the water vapor absorption at 940 nm decrease significantly with the decrease of PWV in the atmosphere, but the water vapor absorption near 1370 nm is still strong. Therefore, we suppose that the water vapor band near 1370 nm may be more suitable for drier atmosphere, which was proved with simulated retrievals by radiative transfer modelling.

*4. 5% (relatively) uniform noise throughout the two bands is not realistic. A proper instrument model should be used since SNR will decrease at wavelengths for which the transmitted irradiance is low in the case of a grating spectrometer. This could be why the results are quite different between simulated data (Fig. 5) and real data (Fig. 10).*

**Responses:** Thank you for your comment and sorry for the unclear sentences given in previous manuscript. In the inversion test, the noise imposed on the theoretical spectrum is not a uniform 5% noise, but a random noise within ±5% superimposed on each wavelength. The results in Fig. 5 and Fig. 11 (previous Fig. 10) are consistent, that is, for dry atmosphere, the PWV retrieved by using band near 940 nm are higher than those of the band near 1370 nm. As shown in in Fig. 5, the results of PWV obtained from band near 940 nm are also shown a relatively larger uncertainty due to the noise added in the simulated spectrum.

*Specific comments*

*1. L17: "CE-318" -> "CE-318 sun photometer."*

**Response:** Thank you for your comment. We have corrected it in the revised manuscript (line 21).

*2. Delete "which shows that" and start a new sentence with "The two …"*

**Response:** Thank you for your comment. We have revised it in the manuscript (line 21).

*3. L30: I don't know if acronyms need to be defined in the main body of the paper to make it independent of the abstract.*

**Response:** Thank you for your comment. We checked some articles that such acronyms are allowed.

*4. L31 (and L41): "etc." should not be used like this. "and others" is preferable.*

**Response:** Thank you for your comment. We have corrected it in the revised manuscript (line 32 and line 41).

*5. L41: I suggest deleting "etc." here.*

**Response:** Thank you for your comment. We have revised it in the manuscript (line 41).

*6. L44: Define PHOTONS*

**Response:** Thank you for your comment. We have included the full name of PHOTONS in the manuscript (line 45).

*7. L44: CE-318" -> "the CE-318"*

**Response:** Thank you for your comment. We have revised it in the manuscript (line 46).

*8. L48: Delete "the"*

**Response:** Thank you for your comment. We have revised it in the manuscript (line 50).

*9. L51: This sentence does not give the spatial domain. Is it a global comparison? 4 references are provided at the end of the sentence but it is not clear which one contains the standard deviation of PWV retrieval differences and the overestimation of AOD.*

**Response:** Thank you for your comment. We have described it in detail in the manuscript (lines 52-55). The PWV retrievals difference between PSR and CIMEL is quoted from Table 2 of Kazadzis et al. (2014). The AOD retrievals difference between PSR and CIMEL is quoted from Figure 3 of Kazadzis et al. (2018a) and also described in Kazadzis et al. (2014).

*10. L52: End this long sentence after "CE-318".*

**Response:** Thank you for your comment. We have revised it in the manuscript (line 55).

*11. L52: The second half of this sentence "and the PWV given…" does not make sense. What is meant by* "*integration"? 940 nm is mentioned, so what are the "different wavelengths"?*

**Response:** Thank you for your comment. Sorry for the error in our description, it has been revised in the manuscript (line 57-58).

*12. L52: "single" -> "a single"*

**Response:** Thank you for your comment. This has been corrected it in the revised manuscript.

*13. L63: environment -> environments*

**Response:** Thank you for your comment. We have corrected it in the revised manuscript (line 66).

*14. L63: There should be a sentence stating whether and why 1370 nm would be less useful in humid environments.*

**Response:** Thank you for your comment. We have added some explanations to the manuscript (lines 64-66).

*15. L88: need -> needs*

**Response:** Thank you for your comment. We have corrected it in the revised manuscript (line 90).

*16. L100: Delete the two consecutive sentences starting with "The light grey"*

**Response:** Thank you for your comment. We have corrected it in the revised manuscript (line 103).

*17. L107: 0 ˚ solar -> 0 ˚ solar*

**Response:** Thank you for your comment. We have revised it in the manuscript (line 106).

*18. L113: atmosphere -> atmospheres*

**Response:** Thank you for your comment. We have revised it in the manuscript (line 112).

*19. L146 (Table 2): mode -> model*

**Response:** Thank you for your comment. We have revised it in the manuscript (line 146).

*20. L146 (Table 2): "Altitude" –> "Altitude of surface"*

**Response:** Thank you for your comment. We have revised it in the manuscript (line 146).

*21. L146 (Table 2): DISORT is not needed for modelling transmittance. DISORT is used when scattered light into the field of view is considered.*

**Response:** Thank you for your comment. We have revised it in the manuscript (line 146).

*22. L151: Give the update equation (i.e. how is the increase/decrease of PWV calculated from iteration to iteration). Is Chahine's method used?*

**Response:** Thank you for your comment. We did not use Chahine's method, and the increase or decrease of PWV depends on whether Δ in Eq. 6 in the manuscript is positive or negative as described in lines 148-150.

*23. L156: The PWV might be randomly generated but it the upper bound is clearly not 0.5 as shown in Fig. 5.*

**Response:** Thank you for your comment. The PWV we input is a random number within 0-0.5 cm, but the curves where the transmittance reaches 0 are filtered out, which leads to the fact that the maximum value of PWV in the Fig.5 does not seem to reach 0.5 cm. We have made revisions in the manuscript (line 154).

*24. L156: "0-0.5" -> "0 and 0.5"*

**Response:** Thank you for your comment. We have revised it in the manuscript (line 154).

*25. L156-157: Delete "and generating 1000 simulated spectral curves."*

**Response:** Thank you for your comment. We have revised it in the manuscript (line 154).

*26. L160: "in" -> "in a"*

**Response:** Thank you for your comment. We have revised it in the manuscript (line 158).

*27. L162 (Fig. 5 and Fig. 10): use grey font for the statistics for Band 1.*

**Response:** Thank you for your comment. We have revised it in the manuscript.

*28. L168: The transmittance due to Rayleigh scattering is approximate. Better formulations exist, but this might be OK since 340 nm seems to be the shortest wavelength used for AOD.*

**Response:** Thank you for your comment. We will pay attention to using the updated Rayleigh scattering formula in our future work.

*29. L171: "the wavelengths used for AOD inversion were carefully selected by using MODTRAN to calculate and filter the wavelengths corresponding to the transmittances greater than 0.999 that do not include Rayleigh scattering and continuous water vapor absorption" -> "the wavelength used for AOD inversion are those for which the MODTRAN transmittances excluding Rayleigh scattering and water vapor absorption are greater than 0.999."*

**Response:** Thank you for your comment. We have revised it in the manuscript (lines 168-169).

*30. L175: "as" –> "as described in"*

**Response:** Thank you for your comment. We have revised it in the manuscript (line 172).

*31. L177 (Fig. 6): 0.99 -> 0.999 (in legend)*

**Response:** Thank you for your comment. We have revised it in the manuscript (line 175).

*32. L184: on -> from*

**Response:** Thank you for your comment. We have revised it in the manuscript (line 203).

*33. L190 (Fig. 7 caption): Either here or preferably in Table 2, the authors should clearly state the average PWV for the "low" or "rich" water content. There is no mention of what 'rich' means quantitatively in this paper.*

**Response:** Thank you for your comment. We have indicated in the abstract that a dry environment refers to PWV<0.5 cm, and other conditions are called rich in water vapor content.

*34. L204: "Presumably due to the large calibration uncertainty…" This explanation is rather simple. Also, the first part of this sentence really belongs at the end of the previous sentence. These two extreme wavelengths have large AOD relative uncertainties for non-instrumental reasons. At 340 nm, aerosols contribute more weakly to the total extinction because of strong Rayleigh scattering and ozone absorption. At 1640 nm, aerosol extinction is weak, particularly when particle size is small (radii < 160 nm). This brings me to another point. Table 2 says there is no boundary layer aerosol used. So how is AOD retrieved? Show the true AOD at each wavelength in Fig. 8.*

**Response:** Thank you for your comment. We have added more explanations in the revised manuscript (lines 221-225). The MODTRAN model settings in Table 2 are only used for PWV inversion. The inversion of AOD is based on Beer Lambert's law that described in sect. 3.3.

*35. L225: The discussion here needs to be improved. The authors' statement should be clear that biases do not result from low signal-to-noise ratio.*

**Response:** Thank you for your comment. We have revised it in the manuscript (lines 243-246).

*36. L228: "with decreasing" -> "for low"*

**Response:** Thank you for your comment. We have revised it in the manuscript (line 247).

*37. L229 (Fig. 10): The 1370 nm band is worse in terms of slope, intercept, and R.*

**Response:** Thank you for your comment. Figure 10 is Fig. 11 in the revised manuscript. Figure 11 is a linear fitting diagram of the PWV retrievals of the EKO instruments and the PWV retrievals of CE-318 sun photometer, and the PWV retrievals of CE-318 cannot be completely considered as the true value. Therefore, the worse fitting effect of the PWV retrievals in the band near 1370 nm and those of CE-

318 in Fig. 11 does not mean that the PWV inversion effect of 1370 nm in a dry environment is worse.

*38. L235: "may be more accurate" is not supported by the evidence (Fig. 10). Speculative statements should be removed even if they are logically defensible.*

**Response:** Thank you for your comment. As discussed in question 37 Fig. 11 can only show that the PWV retrievals of the band near 940 nm are larger than those near 1370 nm, but it cannot come to conclusion which one are more accurate. Therefore, we just introduce the 1370 nm band because the measurements are available, the results shown in Fig. 5 are consistent with Fig. 11, so we suppose 1370 nm could be introduced for PWV retrievals of dry atmosphere when measurements is available.

*39. L247: Why would AOD provide "some assistance" for retrieval of other trace gases from direct sun measurements? L247: "other trace gases retrieval." -> "retrieval of abundances of other trace gases."*

**Response:** Thank you for your comment. We removed such ambiguous statements.

*40. L258. Start a new sentence: "The specific ..."*

**Response:** Thank you for your comment. We have revised it in the manuscript (line 275).

*41. L293: Very questionable speculation…. should be removed unless it can be supported using real data.*

**Response:** Thank you for your comment. We explained the reason for this speculation in the previous responses.

*42. L296: are -> is*

**Response:** Thank you for your comment. We have revised it in the manuscript (line 304).

*43. L298: Start a new sentence after "wavelengths": "This will be considered in the future version.*

**Response:** Thank you for your comment. We have included the CSR correction in this round of revisions and removed this statement in the revised manuscript.

---

## Author Response (AR3)

**Response to Referee #4:**

The authors greatly appreciate Referee #4 for re-reviewing our manuscript and providing valuable comments, which have helped to improve the quality of the paper in both sciences and writing. In the following, we present our responses to comments. The referees' comments are in blue italic and our responses are in black.

**Main comments:**

*I forgot that I had reviewed an earlier draft of this paper and reviewed this version "independently" of the previous one. I find the same issues I had with the previous version. I think the paper is quite good on the whole, but there are a couple of unsolved problems:*

*1) MODTRAN 4.3 is outdated.*

**Responses:** Thank you for your comment. You are right, the current version MODTRAN 4.3 is outdated, and a few spectroscopic data are updated in the new version, such as MODTRAN 5.2. For the relative coarse spectral resolution spectrometer (FWHM=7 nm for MS711 & 712 used in this paper), MODTRAN V4.3 is accurate enough, we compared the water vapor transmittances calculated by MODTRAN5.2 and MODTRAN4.3 as shown in Fig. 1 below, they are very close to each other. Moreover, the inversion algorithm uses the integrated transmittance of a narrow band near 0.94 and 1.37-micrometer stead of a single wavelength, and there is almost no difference for calculations by MODTRAN4.3 or MODTRAN5.2.

[Figure]

**Figure. 1 Water vapor transmittances calculated by MODTRAN4.3 and MODTRAN5.2 under the same conditions, respectively (SZA=0 °, PWV=1.0 g\*m$^{-2}$, FWHM=7 nm).**

*2) Fig. 11 shows that the slope, intercept and correlation coefficient are all worse for the 1.36 micron band, so the statement on L297 is not supported. Section 6 contains a similar statement.*

**Responses:** In Fig. 11, the PWV retrieved from CE-318 is used as the reference. Compare to the retrievals of CE-318, the PWV retrieved with the band near 940nm (BAND1) shows a little higher than that of the band near 1370nm (BAND2), the results from Band1 are closer to that of CE-318. From this point of view, the results from Band2 seem 'worse'. However, all the retrievals cannot be taken as absolute true values. Through the radiative transfer model, the simulated retrievals show that (Fig. 5), the PWV retrieved from BAND2 is closer to the input which is assumed to be the "True" value. Therefore, we proposed that, for a dry atmosphere, a stronger water vapor band around 1370 nm can try to be introduced for PWV inversion if measurements are available. But the statement "the retrievals from Band2 are more

accurate" is not an accurate description, and we have revised it in the manuscript. (Lines 254-257 and Lines 299-302)

*Specific comments*

*1. L37: "economical to build observation network" -> "it is economical to build an observation network with them"*

**Responses:** Thanks, we have revised it in the manuscript. (Line 37)

*2. L55: "940nm" -> "940 nm"*

**Responses:** Thanks, we have revised it in the manuscript. (Line 55)

*3. L61: Why is the "three-parameter formulation method" very sensitive to 'air quality'? I don't believe this is true since path lengths are geometric. Also "formulation method" is redundant in my opinion.*

**Responses:** Thank you for your comment, and you are right, it should be 'air mass', we have corrected it in the revised manuscript. (Line 61)

*4. L65: "easily not" -> "not easily"*

**Responses:** Thanks, we have revised it in the manuscript. (Line 65)

*5. L78: 10-15 -> 3*

**Responses:** Thank you for your comment. We checked the manual and data again, you are right, the CE-318 can be set to run every 3minute, but normally it runs every 10–15 min. (Line 78)

*6. L79: resolution is the wrong word. 'bandwidth' is better. Are these values true for both UV bands and for the three visible bands and for the 4 near-IR bands?*

**Responses:** Thank you for your comment. We have revised it in the manuscript. (Lines 79-80)

*7. L86: are -> is*

**Responses:** Thanks, we have corrected it in the revised manuscript. (Line 85)

*8. L90: Considering -> Consider*

**Responses:** Thanks, We have corrected it in the revised manuscript. (Line 89)

*9. L91: are -> is*

**Responses:** Thank you for your comment. We have corrected it in the revised manuscript. (Line 90)

*10. L130: 'completely' is an adverb and does not belong here.*

**Responses:** Thank you for your comment. We have revised it in the manuscript. (Line 129)

*11. L160: Note that random noise cannot give a biased slope, non-zero intercept or any MB. This is simply a point of information, no need to change the wording.*

**Responses:** Thanks for the informative comments.

*12. L169: great -> greater*

**Responses:** Thank you for your comment. We have revised it in the manuscript. (Line 168)

*13. L176: extenuated -> attenuated*

**Responses:** Thank you for your comment. We have corrected it in the revised manuscript. (Line 175)

*14. L194: "under high aerosol loading atmosphere" -> "in an aerosol-laden environment"*

**Responses:** Thank you for your comment. We have revised it in the manuscript. (Line 193)

*15. L223: "above" -> "the above"*

**Responses:** Thank you for your comment. We have corrected it in the revised manuscript. (Line 222)

*16. L255: atmosphere -> conditions*

**Responses:** Thank you for your comment. We have revised it in the manuscript. (Line 254)

*17. L257: atmosphere -> periods (or conditions)*

**Responses:** Thank you for your comment. We have made revisions to the manuscript. (Line 256)

*18. L295: "at the" -> "at"*

**Responses:** Thank you for your comment. We have corrected it in the revised manuscript. (Line 295)

*19. L296: "near-infrared band" -> "other near-infrared bands"*

**Responses:** Thank you for your comment. We have made revisions to the manuscript. (Line 296)

*20. L299: "a" -> "a dry"*

**Responses:** Thank you for your comment. We have corrected it in the revised manuscript. (Line 299)

*21. L300: This conclusion is not supported by the results. Please reword. Maybe consider limiting to even drier conditions.*

**Responses:** Thank you for your comment. We have rewritten this description as. (Lines 297-302)

"The PWV retrieved from EKO instruments and CE-318 at the band near 940 nm are in good agreement…….. However, under dry conditions, there is little difference between the retrieved PWV from BAND1 (around 940 nm) and BAND2 (around 1370 nm), simulations through radiative transfer modelling show that the retrieved PWV with band 1370 nm is closer to the "true"

value. Therefore, we proposed that a stronger water vapor band near 1370nm can be introduced for PWV retrieval at the dry atmospheres in case of the measurements are available. "

---

## Author Response (AR4)

**Response to Associate Editor:**

The authors greatly appreciate the Associate Editor for the review of submitted manuscript and the Editorial Office for their assistance during the manuscript submission and revision process.